# Detecting Abrupt Changes in Sequential Pairwise Comparison Data

**Wanshan Li**
Department of Statistics & Data Science
Carnegie Mellon University
wanshanl@andrew.cmu.edu

**Daren Wang**
Department of ACMS
University of Notre Dame
dwang24@nd.edu

**Alessandro Rinaldo**
Department of Statistics & Data Science
Carnegie Mellon University
arinaldo@cmu.edu

## Abstract

The Bradley-Terry-Luce (BTL) model is a classic and very popular statistical approach for eliciting a global ranking among a collection of items using pairwise comparison data. In applications in which the comparison outcomes are observed as a time series, it is often the case that data are non-stationary, in the sense that the true underlying ranking changes over time. In this paper we are concerned with localizing the change points in a high-dimensional BTL model with piecewise constant parameters. We propose novel and practicable algorithms based on dynamic programming that can consistently estimate the unknown locations of the change points. We provide consistency rates for our methodology that depend explicitly on the model parameters, the temporal spacing between two consecutive change points and the magnitude of the change. We corroborate our findings with extensive numerical experiments and a real-life example.

## 1 Introduction

Pairwise comparison data are among the most common types of data collected for the purpose of eliciting a global ranking among a collection of items or teams. The Bradley-Terry-Luce model (Bradley and Terry, 1952; Luce, 1959) is a classical and popular parametric approach to model pairwise comparison data and to obtain an estimate of the underlying ranking. The Bradley-Terry-Luce model and its variants have been proven to be powerful approaches in many applications, including sports analytics (Fahrmeir and Tutz, 1994; Masarotto and Varin, 2012; Cattelan et al., 2013), bibliometrics (Stigler, 1994; Varin et al., 2016), search analytics (Radlinski and Joachims, 2007; Agresti, 2013), and much more.

To introduce the BTL model, suppose that we are interested in ranking $n$ distinct items, each with a (fixed but unobserved) positive preference score $w_i$, $i \in [n]$, quantifying its propensity to beat other items in a pairwise comparison. The BTL model assumes that the outcomes of the comparisons between different pairs are independent Bernoulli random variables such that, for a given pair of items, say $i$ and $j$ in $[n] := \{1, \ldots, n\}$, the probability that $i$ is preferred to (or beats) $j$ is

$$P_{ij} = \mathbb{P}\left(i \text{ beats } j\right) = \frac{w_i^*}{w_i^* + w_j^*}, \ \forall \ i, j \in [n]. \tag{1.1}$$

A common reparametrization is to set $w_i^* = \exp(\theta_i^*)$ for each $i$, where $\boldsymbol{\theta}^* := (\theta_1^*, \ldots, \theta_n^*)^\top \in \mathbb{R}^n$. To ensure identifiability it is further assumed that $\sum_{i \in [n]} \theta_i^* = 0$.

36th Conference on Neural Information Processing Systems (NeurIPS 2022).

The properties and performance of the BTL model have been thoroughly studied under the assumption that the outcomes of all the pairwise comparisons are simultaneously available and follow the same BTL model. In many applications however, it is very common to observe pairwise comparison data sequentially (i.e. one at a time), with time stamps over multiple time periods. In these cases, it is unrealistic to assume that observations with different time stamps come from the same distribution. For instance, in sports analytics, the performance of teams often changes across match rounds, and Fahrmeir and Tutz (1994) utilized a state-space generalization of the BTL model to analyze sport tournaments data. Ranking analysis with temporal variants has also become increasingly important because of the growing needs for models and methods to handle time-dependent data. A series of results in this direction can be found in Glickman (1993), Glickman and Stern (1998), Cattelan et al. (2013), Lopez et al. (2018), Maystre et al. (2019), Bong et al. (2020), Karlé and Tyagi (2021) and references therein. Much of the aforementioned literature on time-varying BTL model postulates that temporal changes in the model parameters are smooth functions of time and thus occur gradually on a relatively large time scale. However, there are instances in which it may be desirable to instead model abrupt changes in the underlying parameters and estimate the times at which such change has occurred. These change point settings, which, to the best of our knowledge, have not been considered in the literature, and are the focus of this paper.

**Contributions**

We make the following methodological and theoretical contributions.

• **Novel change point methodology.** We develop a computationally efficient methodology to consistently estimate the change points for a time-varying BTL model with piece-wise constant parameters. Our baseline procedure Algorithm 1 consists of a penalized maximum likelihood estimator of the BTL model under an $\ell_0$ penalty, and can be efficiently implemented via dynamic programming. We further propose a slightly more computationally expensive two-step procedure in Algorithm 2 that takes as input the estimator returned by our baseline procedure and delivers a more precise estimator with provably better error rates. We demonstrate through simulations and a real life example the performance and practicality of the procedure we develop.

• **Theoretical guarantees.** We obtain finite sample error rates for our procedures that depend explicitly on all the parameters at play: the dynamic range of the BTL model and the number of items to be compared, the number of change points, the smallest distance between two consecutive change points and the minimal magnitude of the difference between the model parameters at two consecutive change points. Importantly, our theory allows for general connected *comparison graph* and it explicitly captures the effect the topology of the comparison graph. Our results hold provided that a critical signal-to-noise ratio condition involving all the relevant parameters is satisfied. We conjecture that this condition is optimal in an information theoretic sense. Both the signal-to-noise ratio condition and the localization rates we obtain exhibit a quadratic dependence on the number of items to be compared, which matches the sample complexity bound for two sample testing for the BTL model recently derived by Rastogi et al. (2020).

We emphasize that the change point setting we consider have not been previously studied and both our methodology and the corresponding theoretical guarantees appear to be the first contribution of its kind in this line of work.

**Related work**

Change point detection is a classical problem in statistics that dates back to 1940s (Wald, 1945; Page, 1954). Contributions in the 1980s established asymptotic theory for change point detection methods (Vostrikova, 1981; James et al., 1987; Yao and Au, 1989). Most of the classical literature studied the univariate mean model. Recently with more advanced theoretical tools developed in modern statistics, more delicate analysis of change point detection came out in high-dimensional mean models (Jirak, 2015; Aston and Kirch, 2018; Wang and Samworth, 2018), covariance models (Aue et al., 2009; Avanesov and Buzun, 2018; Wang et al., 2021b), high-dimensional regression models (Rinaldo et al., 2021; Wang et al., 2021c), network models (Wang et al., 2021a), and temporally-correlated times series (Cho and Fryzlewicz, 2015; Preuss et al., 2015; Chen et al., 2021; Wang and Zhao, 2022).

Although change point detection has already been extensively studied in many different settings, little is known about the case of pairwise comparison data. Höhle (2010) numerically study the CUSUM

method for online change point detection in logit models and BTL models without giving theoretical guarantees. We aim to fill the gap in the literature and propose a theoretically trackable approach that can optimally localize abrupt changes in the pairwise comparison data.

## 2  Model and assumptions

Below we introduce the time-varying BTL model with piece-wise constant coefficients that we are going to study and the sampling scheme for collecting pairwise comparison data over time.

Suppose there is a connected comparison graph $\mathcal{G} = \mathcal{G}([n], E)$ with edge set $E \subseteq E_{\text{full}} := \{(i, j) : 1 \leq i < j \leq n\}$. We assume throughout that data are collected as a time series indexed by $t \in [T] := \{1, \ldots, T\}$ that, at each time point $t$, a single pairwise comparison among a collection of $n$ items is observed. The distinct pair $(i_t, j_t) \in [n]^2$ of items to be compared at time $t$ is randomly chosen from the edge set $E$ of $\mathcal{G}$, independently over time. That is,

$$\mathbb{P}(i_t = i, j_t = j) = \frac{1}{|E|}, \ \forall (i, j) \in E. \tag{2.1}$$

For each $t$, let $y_t \in \{0, 1\}$ denote the outcome of the comparison between $i_t$ and $j_t$, where $y_t = 1$ indicates that $i_t$ beats $j_t$ in the comparison. We assume that $y_t$ follows the BTL model (1.1), i.e.

$$\mathbb{P}_{\boldsymbol{\theta}^*(t)}[y_t = 1|(i_t, j_t)] = \frac{e^{\theta_{i_t}^*(t)}}{e^{\theta_{i_t}^*(t)} + e^{\theta_{j_t}^*(t)}}, \tag{2.2}$$

where $\boldsymbol{\theta}^*(t) = (\theta_1^*(t), \ldots \theta_n^*(t))$ is, a possibly time-varying, parameter that belongs to the set

$$\Theta_B := \{\boldsymbol{\theta} \in \mathbb{R}^n : \mathbf{1}_n^\top \boldsymbol{\theta} = 0, \ \|\boldsymbol{\theta}\|_\infty \leq B\}, \tag{2.3}$$

for some $B > 0$. In the recent literature on the BTL model, the parameter $B$ is referred to as the *dynamic range* (see, e.g., Chen et al., 2019) which readily implies a bound on the smallest possible probability that an item is beaten by any other item. Indeed, it follows from (2.2) and (2.3) that

$$\min_{t \in [T], i, j \in [n]} P_{ij}(t) \geq e^{-2B}/(1 + e^{-2B}) := p_{lb} > 0. \tag{2.4}$$

*Remark* 1. The quantity $p_{lb}$ have appeared in several equivalent forms in the BTL literature, e.g., $\max_{i, j \in [n]} \frac{w_i^*}{w_j^*}$ (Simons and Yao, 1999; Negahban et al., 2017) and $e^{2B}$ (Li et al., 2022). The minimal winning probability $p_{lb}$ provides a way of quantifying the difficulty in estimating the model parameters, with a small $p_{lb}$ implying that some items are systematically better than others, a fact that is known to lead to non-existence of the MLE (see, e.g. Ford, 1957) and to hinder parameter estimability. In the BTL literature the dynamic range $B$ and, as a result, the quantity $p_{lb}$ are often treated as known constants and thus omitted (Shah et al., 2016; Chen et al., 2020), a strong assumption that results in an implicit regularization but potentially hides an important feature of the model. As argued in Bong and Rinaldo (2022), in high-dimensional settings this may not be realistic. We will allow for the possibility of a varying $B$ and $p_{lb}$, and keep track of the effect of these parameters on our consistency rates.

It is convenient to rewrite (2.2) in a different but equivalent form that is reminiscent of logistic regression and will facilitate our analysis. One can express the fact that, at time $t$, the items $i_t$ and $j_t$ are randomly selected from $\mathcal{G}([n], E)$ to be compared using a random $n$-dimensional vector $\mathbf{x}(t)$ that is drawn from the sets of all vectors in $\{-1, 0, 1\}^n$ with exactly two-non-zero entries of opposite sign, namely $x_{i_t}(t) = 1$ and $x_{j_t}(t) = -1$ for $(i_t, j_t) \in E$. Then equation (2.2) can be written as

$$\mathbb{P}_{\boldsymbol{\theta}^*(t)}[y_t = 1|\mathbf{x}(t)] = \psi\left(\mathbf{x}(t)^\top \boldsymbol{\theta}^*(t)\right), \tag{2.5}$$

where $\psi(x) = \frac{1}{1 + e^{-x}}$ is the sigmoid function. For any time interval $\mathcal{I} \subset [T]$ we then assume that the data take the form of an i.i.d. sequence $\{(\mathbf{x}(t), y_t)\}_{t \in \mathcal{I}}$, where each $\mathbf{x}(t)$ is an i.i.d. draw from $\{-1, 0, 1\}^n$ with aforementioned properties and, conditionally on $\mathbf{x}(t)$, $y_t$ is a Bernoulli random variable with success probability (2.2). The negative log-likelihood of the data is then given by

$$L(\boldsymbol{\theta}, \mathcal{I}) = \sum_{t \in \mathcal{I}} \ell_t(\boldsymbol{\theta}), \text{ where } \ell_t(\boldsymbol{\theta}) := \ell(\boldsymbol{\theta}; y_t, \mathbf{x}(t)) = -y_t \mathbf{x}(t)^\top \boldsymbol{\theta} + \log[1 + \exp(\mathbf{x}(t)^\top \boldsymbol{\theta})]. \tag{2.6}$$

For a time interval $\mathcal{I}$, we can define a random *comparison graph* $\mathcal{G}_{\mathcal{I}}(V_{\mathcal{I}}, E_{\mathcal{I}})$ with vertex set $V := [n]$ and edge set $E_{\mathcal{I}} := \{(i,j) : i \text{ and } j \text{ are compared in } \mathcal{I}\}$. It is well-known that the topology of $\mathcal{G}_{\mathcal{I}}(V_{\mathcal{I}}, E_{\mathcal{I}})$ plays an important role in the estimation of BTL parameters (Shah et al., 2016). Under assumption (2.1), the comparison graph over $\mathcal{I}$ follows the random graph model $G([n], |\mathcal{I}|)$, which has $|\mathcal{I}|$ edges randomly picked from the edge set $E$ with replacement. Therefore, the process $\{(\mathbf{x}(t), y_t)\}_{t \in \mathcal{I}}$ is stationary as long as $\boldsymbol{\theta}^*(t)$ is unchanged over $\mathcal{I}$.

In the change point BTL model we assume that, for some unknown integer $K \geq 1$, there exist $K+2$ points $\{\eta_k\}_{k=0}^{K+1}$ such that $1 = \eta_0 < \eta_1 < \cdots < \eta_K < \eta_{K+1} = T$ and $\boldsymbol{\theta}^*(t) \neq \boldsymbol{\theta}^*(t-1)$ whenever $t \in \{\eta_k\}_{k \in [K]}$. Define the *minimal spacing* $\Delta$ between consecutive change points and the *minimal jump size* $\kappa$ as

$$\Delta = \min_{k \in [K+1]} (\eta_k - \eta_{k-1}), \quad \kappa = \min_{k \in [K+1]} \|\boldsymbol{\theta}^*(\eta_k) - \boldsymbol{\theta}^*(\eta_{k-1})\|_2. \tag{2.7}$$

As we mentioned in the introduction, the goal of change point localization is to produce an estimator of the change points $\{\hat{\eta}_k\}_{k \in [\hat{K}]}$ such that, with high-probability as $T \to \infty$, we recover the correct number of change points and the localization error is a vanishing fraction of the minimal distance between change points, i.e. that

$$\hat{K} = K, \text{ and } \max_{k \in [K]} |\hat{\eta}_k - \eta_k|/\Delta = o(1). \tag{2.8}$$

In change point literature, estimators satisfying the above conditions are called *consistent*. In the next section we will present two change point estimators and prove their consistency.

## 3   Main results

To estimate the change points, we solve the following regularized maximum likelihood problem over all possible partitions $\mathcal{P}$ of the time course $[T]$:

$$\hat{\mathcal{P}} = \arg\min_{\mathcal{P}} \left\{ \sum_{\mathcal{I} \in \mathcal{P}} L(\hat{\boldsymbol{\theta}}(\mathcal{I}), \mathcal{I}) + \gamma |\mathcal{P}| \right\}, \quad \hat{\boldsymbol{\theta}}(\mathcal{I}) = \arg\min_{\boldsymbol{\theta} \in \Theta_B} L(\boldsymbol{\theta}, \mathcal{I}), \tag{3.1}$$

where $L(\boldsymbol{\theta}, \mathcal{I})$ is the negative log-likelihood function for the BTL model defined in (2.6) and $\gamma > 0$ is an user-specified tuning parameter. Here a partition $\mathcal{P}$ is defined as a set of integer intervals:

$$\mathcal{P} = \{[1, p_1), [p_1, p_2), \ldots, [p_{K_{\mathcal{P}}}, T]\}, 1 < p_1 < p_2 < \cdots < p_{K_{\mathcal{P}}} < T. \tag{3.2}$$

With $\tilde{K} = K_{\hat{\mathcal{P}}} = |\hat{\mathcal{P}}| - 1$, the estimated change points $\{\tilde{\eta}_k\}_{k \in \tilde{K}}$ are then induced by $\tilde{\eta}_k = \hat{p}_k$, $k \in [\tilde{K}]$. The optimization problem (3.1) has an $\ell_0$-penalty, and can be solved by a dynamic programming algorithm described in Algorithm 1 with $O(T^2 \mathcal{C}(T))$ complexity (Friedrich et al., 2008; Rinaldo et al., 2021), where $\mathcal{C}(T)$ is the complexity of solving $\min_{\boldsymbol{\theta}} L(\boldsymbol{\theta}, [1, T])$.

In this section, we will demonstrate that the estimator returned by Algorithm 1 is consistent. Towards that goal, we require the following signal-to-noise ratio condition involving the parameters $\Delta$, $\kappa$, $B$, $n$, the sample size $T$, and the topological property of the underlying comparison graph $\mathcal{G}([n], E)$.

**Assumption 3.1** (Signal-to-noise ratio). Let $\{(\mathbf{x}(t), y_t)\}_{t \in [T]}$ be i.i.d. observations generated from model (2.1) and (2.5) with parameters $\{\boldsymbol{\theta}^*(t)\} \subset \Theta_B$ defined in (2.3). We assume that for a diverging sequence $\{\mathcal{B}_T\}_{T \in \mathbb{Z}^+}$,

$$\Delta \cdot \kappa^2 \geq \mathcal{B}_T p_{lb}^{-4} K \frac{|E| n d_{\max}}{\lambda_2^2(L_{\mathcal{G}})} \log(Tn), \tag{3.3}$$

where we recall that $p_{lb} := \frac{e^{-2B}}{1 + e^{-2B}}$, $d_{\max}$ is the maximal degree of nodes in $\mathcal{G}$ and $\lambda_2(L_{\mathcal{G}})$ is the second smallest eigenvalue of the Laplacian of $\mathcal{G}$ [1].

The formulation of signal-to-noise ratio conditions involving all the parameters of the model has become a staple of modern change point analysis literature. To provide some intuition, the term $\Delta \cdot \kappa^2$ is a proxy for the strength of the signal of change points in the sense that the localization and

---

[1]For a simple undirected graph $\mathcal{G}$ with (binary) adjacency matrix $A$, the Laplacian $L_{\mathcal{G}} := D - A$ where $D = \text{diag}(d_1, \cdots, d_n)$ where $d_i$ is the degree of node $i$.

detection problems are expected to become easier, as the magnitude of the jumps and the spacing between change points increase. On the other hand, the right hand side of Equation (3.3) collects terms that impact negatively the difficulty of the problem: the smaller the minimal win probability $p_{lb}$ and the algebraic connectivity $\lambda_2(L_\mathcal{G})$, the larger the number of items $n$ to compare and the number of change points $K$, the more difficult it is to estimate the change points.

*Remark* 2 (One the topology of $\mathcal{G}$). When the comparison graph $\mathcal{G}$ is a *complete graph*, we have $|E| = \frac{n(n-1)}{2}$, $d_{\max} = n - 1$, $\lambda_2(L_\mathcal{G}) = n$, so the assumption becomes

$$\Delta \cdot \kappa^2 \geq \mathcal{B}_T p_{lb}^{-4} K n^2 \log(Tn). \tag{3.4}$$

In this case, the comparison graph $\mathcal{G}_\mathcal{I}([n], E_\mathcal{I})$ is random graph $G(n, m)$ that have $m$ edges sampled uniformly randomly with replacement. $G(n, m)$ is similar to an Erdös-Rényi graph that is commonly used in the ranking literature (Chen et al., 2019, 2020). In this regard, our result, which directly reflects the impact of the general topology of the sampling graph, is fairly general and in line with recent advances in statistical ranking.

Also note that in general, $\lambda_2 \leq \lambda_n \leq 2d_{\max}$, so the assumption (3.3) ensures that the sample complexity $m \geq C_0 \frac{|E| \log n}{\lambda_2(L_\mathcal{G})}$ in Lemma B.15 is satisfied in the worst case $\kappa^2 \asymp n$.

*Remark* 3 (On the sharpness of the signal-to-noise ratio condition). We will now argue that the requirement (3.1) imposed by the signal-to-noise ratio (SNR for brevity) is reasonably sharp by relating it to the sample complexity of a two-sample testing problem. To that effect, consider the simplified setting in which there is only one change point at time $\Delta = T/2$ and $\mathcal{G}$ is a complete graph. In this case, it can be shown that the SNR condition (3.1) becomes (see Proposition B.5)

$$\Delta \cdot \kappa^2 \geq \mathcal{B}_T p_{lb}^{-2} n^2 \log(Tn), \tag{3.5}$$

i.e. the dependence on the dynamic range $B$ is through $p_{lb}^{-2}$ instead of $p_{lb}^{-4}$. It stands to reason that estimating the unknown change point $\Delta$ should be at least as hard as testing the null hypothesis that there exists a change point at time $\Delta$. Indeed, this testing problem should be easier because $\Delta$ has been revealed and because, in general, testing is easier than estimation. This can in turn be cast as a two-sample testing problem of the form

$$H_0 : \mathbf{P}(\boldsymbol{\theta}^{(1)}) = \mathbf{P}(\boldsymbol{\theta}^{(2)}) \text{ v.s. } H_1 : \frac{1}{n}\|\mathbf{P}(\boldsymbol{\theta}^{(1)}) - \mathbf{P}(\boldsymbol{\theta}^{(2)})\|_F \geq \epsilon, \tag{3.6}$$

where $\epsilon > 0$ is to be specified, $\boldsymbol{\theta}^{(1)}$ and $\boldsymbol{\theta}^{(2)}$ are the BTL model parameters for the first and the last $\Delta$ observations respectively and, for $i \in \{1, 2\}$, $\mathbf{P}(\boldsymbol{\theta}^{(i)})$ is the $n \times n$ matrix of winning probabilities corresponding to the BTL model parameter $\boldsymbol{\theta}^{(i)}$ as specified by (2.2). To see how one arrives at (3.6), we have that, by Proposition B.4,

$$\|\mathbf{P}(\boldsymbol{\theta}^{(1)}) - \mathbf{P}(\boldsymbol{\theta}^{(2)})\|_F^2 \geq \frac{np_{lb}^2}{16}\|\boldsymbol{\theta}^{(1)} - \boldsymbol{\theta}^{(2)}\|_2^2. \tag{3.7}$$

Thus, a change point setting with $\|\boldsymbol{\theta}^{(1)} - \boldsymbol{\theta}^{(2)}\|_2^2 = \kappa^2$, translates into the testing problem (3.6) with $\epsilon^2 = \kappa^2 p_{lb}^2/(16n)$. By Theorem 7 of Rastogi et al. (2020), there exists an algorithm that will return a consistent test for (3.6) based on two independent samples of size $N$ if $N \geq cn^2 \log(n)\frac{1}{n\epsilon^2}$. When we apply this result to the simplified change point settings described above (by replacing $N$ and $\epsilon^2$ with $\Delta$ and $\kappa^2 p_{lb}^2/(16n)$ respectively) we conclude that the sample complexity bound of Theorem 7 of Rastogi et al. (2020) corresponds, up to constants, to the above SNR condition (3.5) save for the terms $\log(T)$ and $\mathcal{B}_T$. Thus, we conclude that the assumed SNR condition for change point localization is essentially equivalent to the sample complexity needed to tackle the simpler two-sample testing problem, an indication that our assumption is sharp.

Finally, we take notice that, when there are multiple change points, in our analysis it appears necessary to strengthen the signal-to-noise ratio condition (3.5) to (3.1) by requiring a dependence on $p_{lb}^{-4}$.

We are now ready to present our first consistency result.

**Theorem 3.2.** *Let* $\{\tilde{\eta}_k\}_{k \in [\tilde{K}]}$ *be the estimates of change points from Algorithm 1 with the tuning parameter* $\gamma = C_\gamma p_{lb}^{-2}(K+1)\frac{nd_{\max}}{\lambda_2(L_\mathcal{G})}\log(Tn)$ *where* $C_\gamma$ *is a universal constant. Under Assumption 3.1 we have*

$$\mathbb{P}\left\{\tilde{K} = K, \quad \max_{k \in [K]}|\tilde{\eta}_k - \eta_k| \leq C_P p_{lb}^{-4} K\frac{|E|nd_{\max}}{\kappa^2\lambda_2^2(L_\mathcal{G})}\log(Tn)\right\} \geq 1 - 2(Tn)^{-2}, \tag{3.8}$$

*where* $C_P > 0$ *is a universal constant that depends on* $C_\gamma$.

Theorem 3.2 gives a high-probability upper bound for the localization error of the output $\{\tilde{\eta}_k\}_{k\in[\tilde{K}]}$ of Algorithm 1. By Assumption 3.1, it follows that as $T \to \infty$, with high probability,

$$\max_{k\in[K]} |\tilde{\eta}_k - \eta_k| \leq C_P p_{lb}^{-4} K \frac{|E| n d_{\max}}{\kappa^2 \lambda_2^2(L_{\mathcal{G}})} \log(Tn) \leq C_P \frac{\Delta}{\mathcal{B}_T} = o(\Delta), \tag{3.9}$$

where we use the singal-to-noise ratio assumption $\Delta \cdot \kappa^2 \geq \mathcal{B}_T p_{lb}^{-4} K \frac{|E| n d_{\max}}{\lambda_2^2(L_{\mathcal{G}})} \log(Tn)$ in the last inequality and the fact that $\mathcal{B}_T$ diverges in the final step. This implies that the estimators $\{\tilde{\eta}_k\}_{k\in[\tilde{K}]}$ are consistent. Moreover, when $K = 0$ or there is no change point, it is guaranteed that, with high probability, Algorithm 1 will return an empty set. We summarize this property as Proposition B.6 and include it in Appendix B.2 due to the limit of space.

---

**Algorithm 1:** Dynamic Programming. DP $(\{(\mathbf{x}(t), y_t)\}_{t\in[T]}, \gamma)$

**INPUT:** Data $\{(\mathbf{x}(t), y_t)\}_{t\in[T]}$, tuning parameter $\gamma$.
Set $S = \emptyset$, $\mathfrak{p} = -\mathbf{1}_T$, $\mathbf{b} = (\gamma, \infty, \ldots, \infty) \in \mathbb{R}^T$. Denote $b_i$ to be the $i$-th entry of $\mathbf{b}$.
**for** $r$ *in* $\{2, \ldots, T\}$ **do**
    **for** $l$ *in* $\{1, \ldots, r-1\}$ **do**
$$b \leftarrow b_l + \gamma + L(\hat{\boldsymbol{\theta}}(\mathcal{I}), \mathcal{I}) \quad \text{where} \quad \mathcal{I} = (l, \ldots, r];$$
        **if** $b < b_r$ **then**
            $b_r \leftarrow b$; $\mathfrak{p}_r \leftarrow l$.

To compute the change point estimates from $\mathfrak{p} \in \mathbb{N}^T$, $k \leftarrow T$.
**while** $k > 1$ **do**
    $h \leftarrow \mathfrak{p}_k$ ; $S = S \cup h$; $k \leftarrow h$.
**OUTPUT:** The estimated change points $S = \{\tilde{\eta}_k\}_{k\in\tilde{K}}$.

---

**Algorithm 2:** Local Refinement.

**INPUT:** Data $\{(\mathbf{x}(t), y_t)\}_{t\in[T]}$, $\{\tilde{\eta}_k\}_{k\in[\tilde{K}]}$, $(\tilde{\eta}_0, \tilde{\eta}_{\tilde{K}+1}) \leftarrow (1, T)$.
**for** $k = 1, \ldots, \tilde{K}$ **do**

$$(s_k, e_k) \leftarrow (2\tilde{\eta}_{k-1}/3 + \tilde{\eta}_k/3, \ \tilde{\eta}_k/3 + 2\tilde{\eta}_{k+1}/3);$$

$$\hat{\eta}_k \leftarrow \operatorname*{arg\,min}_{\eta \in \{s_k+1, \ldots, e_k-1\}} \left\{ \min_{\boldsymbol{\theta}^{(1)} \in \Theta_B} \sum_{t=s_k+1}^{\eta} \ell_t(\boldsymbol{\theta}^{(1)}) + \min_{\boldsymbol{\theta}^{(2)} \in \Theta_B} \sum_{t=\eta+1}^{e_k} \ell_t(\boldsymbol{\theta}^{(2)}) \right\};$$

$$\tag{3.10}$$

**OUTPUT:** $\{\hat{\eta}_k\}_{k\in[\tilde{K}]}$.

---

Inspired by previous works (Wang et al., 2021a; Rinaldo et al., 2021), we can further improve the localization error by applying a local refinement procedure as described in Algorithm 2 to $\{\tilde{\eta}_k\}_{k\in[\tilde{K}]}$. This methodology takes as input any preliminary estimator of the change points that estimates the number of change points correctly with a localization error that is a (not necessarily vanishing) fraction of the minimal spacing $\Delta$, and returns a new estimator with a provably smaller localization error. A natural preliminary estimator is the one returned in Algorithm 1. The next result derives the improved localization rates delivered by the local refinement step. The two improvements are the elimination of the term $K$ in the rate and a better dependence on $p_{lb}$.

**Theorem 3.3.** *Let* $\{\hat{\eta}_k\}_{k\in[\hat{K}]}$ *be the output of Algorithm 2 with input* $\{\tilde{\eta}_k\}_{k\in[\hat{K}]}$ *returned by Algorithm 1. Under Assumption 3.1, for all sufficiently large* $T$ *we have*

$$\mathbb{P}\left\{\hat{K} = K, \quad \max_{k\in[K]} |\hat{\eta}_k - \eta_k| \leq C_R p_{lb}^{-2} \frac{|E| n d_{\max}}{\kappa^2 \lambda_2^2(L_{\mathcal{G}})} \log(Tn)\right\} \geq 1 - 2(Tn)^{-2}, \tag{3.11}$$

*where* $C_R > 0$ *is a universal constant that depends on* $C_\gamma$.

*Remark* 4. By "sufficiently large $T$" in the theorem statement, we mean that $T$ should be large enough to make $\max_{k \in [K]} |\hat{\eta}_k - \eta_k| \leq \Delta/5$ (see Proposition B.3 in Appendix B for details). Such $T$ exists because of Equation (3.9) and the fact that $\mathcal{B}_T$ is diverging in $T$.

We conjecture that the rate (3.11) resulting from the local refinement procedure is, aside possibly from a logarithmic factor, minimax optimal.

## 4 Experiments

In this section, we study the numerical performance of our newly proposed method based on a combination of dynamic programming with local refinement, which we will refer to as DPLR; see Algorithms 1 and 2. We note that the detection of multiple change points in pairwise comparison data has not been studied before, as Höhle (2010) only focus on single change point detection for pairwise comparison data, so we are not aware of any existing competing methods in the literature. Thus, we develop a potential competitor based on the combination of *Wild Binary Segmentation* (WBS) (Fryzlewicz, 2014), a popular method for univariate change point detection, and the likelihood ratio approach studied in Höhle (2010). We will call this potential competitor WBS-GLR (GLR stands for generalized likelihood ratio). Due to the limit of space, we include the detail of WBS-GLR in Appendix A.1, and results of additional experiments in Appendix A.2, where additional settings are considered. Furthermore, we discuss and compare the performance of two other potential competitors in Appendix A.4.

All of our simulation results show that our proposed method DPLR outperforms WBS-GLR in the sense that DPLR gives more accurate change point estimates with similar running time. Each experiment is run on a virtual machine of Google Colab with Intel(R) Xeon(R) CPU of 2 cores 2.30 GHz and 12GB RAM. All of our reproducible code is openly accessible [2]

**Simulation Settings.** Suppose we have $K$ change points $\{\eta_k\}_{k \in [K]}$ in the sequential pairwise comparison data, with $\eta_0 = 1$. We can use $\boldsymbol{\theta}^*(\eta_k)$ to represent the value of true parameters after the change point $\eta_k$. To begin, we define $\theta_i^*(\eta_0)$ as follows. For $1 < i \leq n$, we set $\theta_i^*(\eta_0) = \theta_1^*(\eta_0) + (i-1)\delta$ with some constant $\delta$. In each experiment, we set $\delta$ first and then set $\theta_1^*(\eta_0)$ to make $\mathbf{1}_n^\top \boldsymbol{\theta}^*(\eta_0) = 0$. For a given $n$, we set $\delta = \frac{1}{n-1}\psi^{-1}(p) = \frac{1}{n-1}\log(\frac{p}{1-p})$ where $\psi^{-1}$ is the inverse function of $\psi$ and $p = 0.9$. Recall that $P_{ij} = \psi(\theta_i - \theta_j)$ is the winning probability, so the value of $\delta$ guarantees that the maximum winning probability is 0.9. We consider three types of changes:

Type I (reverse): $\theta_i^*(\eta_k) = \theta_{n+1-i}^*(\eta_0)$.

Type II (block-reverse): $\theta_i^*(\eta_k) = \theta_{[\frac{n}{2}]+1-i}^*(\eta_0)$ for $i \leq [\frac{n}{2}]$; $\theta_i^*(\eta_k) = \theta_{[\frac{n}{2}]+n+1-i}^*(\eta_0)$ for $i > [\frac{n}{2}]$.

Type III (block exchange): $\theta_i^*(\eta_k) = \theta_{i+[\frac{n}{2}]}^*(\eta_0)$ for $i \leq [\frac{n}{2}]$; $\theta_i^*(\eta_k) = \theta_{i-[\frac{n}{2}]}^*(\eta_0)$ for $i > [\frac{n}{2}]$.

We consider four simulation settings. For each setting, we set the comparison graph $\mathcal{G}([n], E)$ to be the complete graph and $T = (K+1)\Delta$ with true change points located at $\eta_i = i\Delta$ for $i \in [K]$. To describe the true parameter at each change point, we use an ordered tuple. For instance, (I, II, III, I) means that $K = 4$ and the true parameters at $\eta_1, \eta_2, \eta_3, \eta_4$ are determined based on $\boldsymbol{\theta}^*(\eta_0)$ and the change type I, II, III, and I, respectively.

For the constrained MLE in Equation (3.1), we use the function in `sklearn` for fitting the $\ell_2$-penalized logistic regression, as it is well-known that the constrained and the penalized estimators for generalized linear models are equivalent. For both DPLR and WBS-GLR, we use $\lambda = 0.1$. For $M$, the number of random intervals in WBS-GLR, we set it to be 50 as a balance of time and accuracy.

For both methods, we use cross-validation to choose the tuning parameter $\gamma$. Given the sequential pairwise comparison data in each trial, we use samples with odd time indices as training data and even time indices as test data. For each tuning parameter, the method is applied to the training data to get estimates of change points. Then a BTL model is fitted to the test data for each interval determined by the estimated change points. The tuning parameter and the corresponding change point estimators with the minimal test error (negative loglikelihood) are selected. We run 100 trials for each setting.

---

[2]Code repository: `https://github.com/MountLee/CPD_BT`

| | $H(\hat{\eta}, \eta)$ | Time | $\hat{K} < K$ | $\hat{K} = K$ | $\hat{K} > K$ |
|---|---|---|---|---|---|
| | **Setting (i)**   $n = 10, K = 3, \Delta = 500$, Change (I, II, III) | | | | |
| DPLR | 9.2 (9.1) | 49.7s (0.7) | 0 | 100 | 0 |
| WBS-GLR | 15.2 (7.9) | 31.9s (3.9) | 0 | 100 | 0 |
| | **Setting (ii)**   $n = 20, K = 3, \Delta = 800$, Change (I, II, III) | | | | |
| DPLR | 9.0 (9.9) | 118.5s (2.2) | 0 | 100 | 0 |
| WBS-GLR | 240.5 (220.3) | 144.2s (12.5) | 0 | 40 | 60 |
| | **Setting (iii)**   $n = 100, K = 2, \Delta = 1000$, Change (I, II) | | | | |
| DPLR | 13.4 (14.4) | 167.4s (3.3) | 0 | 100 | 0 |
| WBS-GLR | 111.9 (195.6) | 215.9s (17.0) | 0 | 79 | 21 |
| | **Setting (iv)**   $n = 100, K = 3, \Delta = 2000$, Change (I, II, III) | | | | |
| DPLR | 12.4 (12.1) | 402.4s (7.4) | 0 | 100 | 0 |
| WBS-GLR | 412.3 (495.5) | 400.0s (40.9) | 0 | 57 | 43 |

Table 1: Comparison of DPLR and WBS-GLR under four different simulation settings. 100 trials are conducted in each setting. For the localization error and running time (in seconds), the average over 100 trials is shown with standard error in the bracket. The three columns on the right record the number of trials in which $\hat{K} < K$, $\hat{K} = K$, and $\hat{K} > K$ respectively.

**Results.**   To measure the localization errors, we use the Hausdorff distance $H(\{\hat{\eta}_i\}_{i \in [\hat{K}]}, \{\eta_i\}_{i \in [K]})$ between the estimated change points $\{\hat{\eta}_i\}_{i \in [\hat{K}]}$ and the true change points $\{\eta_i\}_{i \in [K]}$. The Hausdorff distance $H(S_1, S_2)$ between two sets of scalars is defined as

$$H(S_1, S_2) = \max\{\sup_{x \in S_1} \inf_{y \in S_2} |x - y|, \sup_{y \in S_2} \inf_{x \in S_1} |x - y|\}. \tag{4.1}$$

The results are summarized in Table 1, where we use $H(\hat{\eta}, \eta)$ to denote the localization error for brevity. As we can see, our proposed method DPLR gives more accurate localization with similar running time compared to the potential competitor WBS-GLR.

## 5   Application: the National Basketball Association games

We study the game records of the National Basketball Association (NBA) [3]. Usually a regular NBA season begins in October and ends in April of the next year, so in what follows, a season is named by the two years it spans over. The original data contains all game records of NBA from season 1946-1947 to season 2015-2016. We focus on a subset of 24 teams founded before 1990 and seasons from season 1980-1981 to season 2015-2016. All code of analysis is available online with the data [4]

We start with an exploratory data analysis and the results show strong evidence for multiple change points [5]. Therefore, we apply our method DPLR to the dataset to locate those change points. We use the samples with odd time indices as training data and even time indices as test data, and use cross-validation to choose the tuning parameter $\gamma$.

To interpret the estimated change points, we fit the BTL model on each subset splitted at change point estimates separately. The result is summarized in Table 2. Several teams show significant jumps in the preference scores and rankings around change points. Apart from this quantitative assessment, the result is also firmly supported by memorable facts in NBA history, and we will name a few here. In 1980s, Celtics was in the "Larry Bird" era with its main and only competitor "Showtime" Lakers. Then starting from 1991, Michael Jordan and Bulls created one of the most famous dynasties in NBA history. 1998 is the year Michael Jordan retired, after which Lakers and Spurs were dominating during 1998-2009 with their famous cores "Shaq and Kobe" and "Twin Towers". The two teams together won 8 champions during these seasons. S2010-S2012 is the well-known "Big 3" era of Heat. Meanwhile, Spurs kept its strong competitiveness under the lead of Timothy Duncan. From 2013, with the arise of super stars Stephen Curry and Klay Thompson, Warriors started to take the lead.

---

[3] https://gist.github.com/masterofpun/2508ab845d53add72d2baf6a0163d968

[4] Code repository: https://github.com/MountLee/CPD_BT

[5] Due to the limit of space, we include these results in Appendix A.3.

| S1980-S1985 | | S1986-S1991m | | S1991m-S1997 | | S1998-S2003 | |
| --- | --- | --- | --- | --- | --- | --- | --- |
| Celtics | 1.1484 | Lakers | 1.1033 | Bulls | 0.9666 | Spurs | 0.8910 |
| 76ers | 0.9851 | Pistons | 0.7696 | Jazz | 0.8618 | Lakers | 0.8744 |
| Bucks | 0.7828 | Celtics | 0.7304 | Knicks | 0.5908 | Kings | 0.6833 |
| Lakers | 0.7779 | Trail Blazers | 0.6848 | Suns | 0.5628 | Mavericks | 0.5087 |
| Nuggets | 0.0789 | Bulls | 0.6647 | Rockets | 0.5032 | Trail Blazers | 0.4899 |
| Trail Blazers | 0.0636 | Jazz | 0.5179 | Spurs | 0.4742 | Jazz | 0.3944 |
| Suns | 0.0636 | Bucks | 0.3474 | Trail Blazers | 0.4176 | Timberwolves | 0.3913 |
| Spurs | 0.0611 | Suns | 0.3472 | Cavaliers | 0.3751 | Pacers | 0.3165 |
| Nets | 0.0215 | Rockets | 0.3156 | Magic | 0.3009 | Hornets | 0.1002 |
| Pistons | -0.0252 | 76ers | 0.2195 | Lakers | 0.2730 | 76ers | 0.0993 |
| Knicks | -0.1333 | Cavaliers | 0.1885 | Pacers | 0.2688 | Suns | 0.0721 |
| Rockets | -0.1950 | Mavericks | 0.1798 | Hornets | 0.2465 | Pistons | 0.0249 |
| Jazz | -0.2926 | Knicks | 0.0583 | Heat | 0.1445 | Bucks | -0.0146 |
| Kings | -0.3104 | Warriors | 0.0441 | Pistons | -0.2028 | Rockets | -0.0525 |
| Mavericks | -0.3104 | Spurs | 0.0035 | Nets | -0.2122 | Knicks | -0.1420 |
| Bulls | -0.3115 | Nuggets | -0.0232 | Warriors | -0.3075 | Heat | -0.1455 |
| Warriors | -0.4330 | Pacers | -0.0237 | Celtics | -0.3288 | Nets | -0.2276 |
| Pacers | -0.5500 | Kings | -0.7006 | Kings | -0.4808 | Magic | -0.2650 |
| Clippers | -0.6443 | Nets | -0.7666 | Clippers | -0.5419 | Celtics | -0.2885 |
| Cavaliers | -0.7771 | Clippers | -0.7788 | Bucks | -0.5864 | Nuggets | -0.4894 |
| Heat | NA | Magic | -0.8969 | Nuggets | -0.6272 | Clippers | -0.6250 |
| Hornets | NA | Timberwolves | -0.9554 | Timberwolves | -0.6570 | Cavaliers | -0.6796 |
| Magic | NA | Heat | -0.9874 | 76ers | -0.8869 | Warriors | -0.7362 |
| Timberwolves | NA | Hornets | -1.0418 | Mavericks | -1.1542 | Bulls | -1.1801 |
| S2004-S2006 | | S2007-S2009 | | S2010-S2012 | | S2013-S2015 | |
| Spurs | 1.0532 | Lakers | 1.0097 | Heat | 0.9909 | Warriors | 1.3617 |
| Suns | 0.9559 | Celtics | 0.8699 | Spurs | 0.8653 | Spurs | 1.2728 |
| Mavericks | 0.9338 | Magic | 0.7741 | Bulls | 0.8292 | Clippers | 0.9909 |
| Pistons | 0.8120 | Cavaliers | 0.7466 | Nuggets | 0.5857 | Rockets | 0.6158 |
| Heat | 0.2713 | Spurs | 0.6270 | Lakers | 0.4922 | Trail Blazers | 0.5501 |
| Rockets | 0.1803 | Mavericks | 0.5686 | Mavericks | 0.4121 | Mavericks | 0.4197 |
| Cavaliers | 0.1510 | Jazz | 0.5169 | Clippers | 0.3413 | Cavaliers | 0.3872 |
| Nuggets | 0.1322 | Nuggets | 0.4751 | Celtics | 0.2901 | Heat | 0.3215 |
| Kings | 0.0542 | Suns | 0.4146 | Knicks | 0.1990 | Pacers | 0.3202 |
| Lakers | 0.0166 | Hornets | 0.3593 | Pacers | 0.1233 | Bulls | 0.2104 |
| Nets | -0.0149 | Rockets | 0.3428 | Rockets | 0.1227 | Hornets | 0.0145 |
| Timberwolves | -0.0566 | Trail Blazers | 0.2750 | Jazz | 0.0167 | Pistons | -0.1710 |
| Clippers | -0.0646 | Bulls | -0.1260 | Trail Blazers | -0.0549 | Suns | -0.1787 |
| Bulls | -0.0680 | Pistons | -0.1821 | Magic | -0.0899 | Jazz | -0.1936 |
| Pacers | -0.0824 | Heat | -0.2939 | Warriors | -0.1402 | Celtics | -0.2037 |
| Jazz | -0.1039 | 76ers | -0.3418 | 76ers | -0.1930 | Nets | -0.3093 |
| Magic | -0.2482 | Warriors | -0.3729 | Bucks | -0.2362 | Nuggets | -0.3140 |
| Warriors | -0.2803 | Pacers | -0.3936 | Suns | -0.3228 | Kings | -0.4066 |
| 76ers | -0.3030 | Bucks | -0.5456 | Nets | -0.4589 | Bucks | -0.4516 |
| Celtics | -0.5144 | Kings | -0.7977 | Hornets | -0.4670 | Timberwolves | -0.6266 |
| Hornets | -0.5641 | Knicks | -0.8568 | Timberwolves | -0.6034 | Magic | -0.6398 |
| Bucks | -0.6555 | Nets | -0.8935 | Kings | -0.6929 | Knicks | -0.6591 |
| Knicks | -0.7101 | Clippers | -1.0853 | Pistons | -0.7807 | Lakers | -0.9431 |
| Trail Blazers | -0.8947 | Timberwolves | -1.0901 | Cavaliers | -1.2285 | 76ers | -1.3676 |

Table 2: Fitted $\hat{\theta}$ (rounded to the fourth decimal) for 24 selected teams in seasons 1980-2016 of the National Basketball Association. Teams are ranked by the MLE $\hat{\theta}$ on subsets splitted at the estimated change points given by our DPLR method. S1980 means season 1980-1981 and S1991m means the middle of season 1991-1992. Heat(1988), Hornets(1988), Magic(1989), and Timberwolves(1989) were founded after S1985, so the corresponding entries are marked as NA.

# 6  Conclusions

We have formulated and investigate a novel change point analysis problem for pairwise comparison data based on a high-dimensional BTL model. We have developed a novel methodology that yields consistent estimators of the change points, and establish theoretical guarantees with nonasymptotic

localization error. To the best of our knowledge, this is the first work in the literature that addresses in both a methodological and theoretically sound way multiple change points in ranking data.

Although we filled a big gap in the literature, there remain many open and interesting problems for future work. First, we only consider pairwise comparison data modeled by the BTL model. Of course, there are other popular ranking models for general ranking data, e.g., the Plackett-Luce model(Luce, 1959; Plackett, 1975), Stochastically Transitive models(Shah et al., 2017), and the Mallows model (Tang, 2019). It would be interesting to see that for those models how different the method and theory would be from our settings. We present some exploratory results on this in Appendix A.4. Second, we have focused on *retrospective* setting of change point detection and *passive* setting of ranking. On the other hand, *online* change point detection (Vovk, 2021) and *active ranking* (Heckel et al., 2019; Ren et al., 2021) are widely used in practice. Thus, it would be interesting to consider the online or active framework in change point detection for ranking data. Third, in the recent change point detection literature, incorporating temporal dependence is of growing interest (Chen et al., 2021; Wang and Zhao, 2022), so investigating how temporal dependence in the pairwise comparison data can affect our results seems like a worthwhile direction.

At last, we discuss potential societal impacts of our work. The BTL model does have applications with potentially undesirable societal impacts, e.g., sports-betting (McHale and Morton, 2011), which could amplify the negative impacts of gambling. We recommend using our method for research purposes rather than gambling-driven purposes.

### Acknowledgments

We would like to thank the anonymous reviewers for their feedback which greatly helped improve our exposition. Wanshan Li and Alessandro Rinaldo acknowledge partial support from NSF grant DMS-EPSRC 2015489.

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
