# Appendix of "Detecting Abrupt Changes in Sequential Pairwise Comparison Data"

This is the appendix of the paper "Detecting Abrupt Changes in Sequential Pairwise Comparison Data" as a supplementary material. It contains two parts:

1. Appendix A for some supplements to numerical results in Sections 4 and 5.

2. Appendix B for the proof of main results and some additional propositions used in the main text.

## A   Appendix: supplementary to numerical results

### A.1   Wild binary segmentation based on likelihood

*Binary segmentation* is a classical and popular method for detecting change points that can at least date back to Scott and Knott (1974). It is based on the so-called CUSUM statistics. In the case where we are interested in detecting the change point in the mean of univariate random variables $\{Y_t\}_{t\in[T]}$, the CUSUM statistic at time $t$ over an interval $(s,e)$ is defined as

$$\text{CUSUM}(t;s,e) := |\sqrt{\frac{e-t}{(e-s)(t-s)}} \sum_{i=s+1}^{t} Y_i - \sqrt{\frac{t-s}{(e-s)(e-t)}} \sum_{i=t+1}^{e} Y_i|. \tag{A.1}$$

It is known that Binary Segmentation is consistent but not optimal (Venkatraman (1992)). As an improvement, Fryzlewicz (2014) propose Wild Binary Segmentation and show that it has a better localization rate.

---

**Algorithm 3:** Wild Binary Segmentation. $\text{WBS}((s,e), \{(\alpha_m,\beta_m)\}_{m\in[M]}, \gamma)$

---

**INPUT:** Independent samples $\{Z_i\}_{i\in[n]}$, collection of intervals $\{(\alpha_m,\beta_m)\}_{m\in[M]}$, tuning parameters $\gamma > 0$.

**for** $m = 1, \ldots, M$ **do**
> $(s_m, e_m) \leftarrow [s,e] \cap [\alpha_m, \beta_m]$
> **if** $e_m - s_m > 1$ **then**
> > $b_m \leftarrow \arg\max_{s_m+1 \leq t \leq e_m-1} \mathcal{R}(t; s_m, e_m)$
> > $a_m \leftarrow \mathcal{R}(b_m; s_m, e_m)$
>
> **else**
> > $a_m \leftarrow -1$

$m^* \leftarrow \arg\max_{m\in[M]} a_m$
**if** $a_{m^*} > \gamma$ **then**
> add $b_{m^*}$ to the set of estimated change points
> $\text{WBS}((s, b_{m*}), \{(\alpha_m,\beta_m)\}_{m\in[M]}, \gamma)$
> $\text{WBS}((b_{m*}+1, e), \{(\alpha_m,\beta_m)\}_{m\in[M]}, \gamma)$

**OUTPUT:** The set of estimated change points.

---

Algorithm 3 shows the general framework of WBS algorithm. For univariate mean, we have $\mathcal{R}(t; s, e) = \text{CUSUM}(t; s, e)$. While for our problem, the Bradley-Terry model, we set $\mathcal{R}(t; s, e)$ to be the (logarithmic) generalized likelihood ratio given by

$$\mathcal{R}(t;s,e) = GLR(t;s,e) := \max_{\boldsymbol{\theta}_l \in \Theta_B} \{-L(\boldsymbol{\theta}_l, [s,t))\} + \max_{\boldsymbol{\theta}_r \in \Theta_B} \{-L(\boldsymbol{\theta}_r, [t,e])\} - L_{s,e}, \tag{A.2}$$

where $L_{s,e} := \max_{\boldsymbol{\theta} \in \Theta_B} \{-L(\boldsymbol{\theta}, [s,e])\}$ and $L(\boldsymbol{\theta}, \mathcal{I})$ is the negative log-likelihood function over interval $\mathcal{I}$, as is defined in Equation (2.6). The use of generalized likelihood ratio in change point detection has been demonstrated in many previous works (Höhle, 2010; Wang et al., 2020). In fact, when $\{Y_t\}_{t\in[T]}$ follows Gaussian distribution with known variance, the GLR statistic at $t$ is the square of $\text{CUSUM}(t; s, e)$.

Similar to the DP approach, WBS also has a tuning parameter $\gamma$. By Equation (A.2) and the design of Algorithm 1 and 3, we know that the $\gamma$ parameters for both DP and WBS-GLR act as the threshold

for the GLR statistic. Therefore, one should use the same candidate list of $\gamma$ for both methods when tuning parameters by cross-validation for fair comparison, as we do in all experiments.

In addition, the number of intervals $M$ acts as another tuning parameter and makes WBS more tricky to apply compared to the DP approach. In practice, people usually set intervals $\{(\alpha_m, \beta_m)\}_{m \in [M]}$ to be uniformly randomly sampled from $[0, T]$. Although it doesn't affect the theoretical guarantee too much Wang et al. (2020), numerically the performance of WBS heavily depends on $M$. Typically, the larger $M$ is, the more accurate the result is, and the more time it takes to execute WBS. When the model of the data is simple, e.g., univariate mean model, computation of $\mathcal{R}(t; s, e)$ is cheap and one can just set $M$ to be large to improve the localization accuracy. However, for more complex models like the BTL model, a large $M$ may not be computationally affordable, so it can be hard to set an appropriate value for $M$.

## A.2    Additional simulated experiments

In Section 4, we consider simulation settings where both the signals $\boldsymbol{\theta}^*(t)$ and changes of $\boldsymbol{\theta}^*(t)$ at change points are set in a deterministic way. In this section, we consider experiments where entries of $\boldsymbol{\theta}^*(t)$ are randomly sampled and are randomly permuted at each change point. Suppose we have $K$ change points $\{\eta_k\}_{k \in [K]}$ in the sequential pairwise comparison data, with $\eta_0 = 1$. We use $\boldsymbol{\theta}^*(\eta_k)$ to represent the value of true parameters after the change point $\eta_k$.

To begin, we set $\{\theta_i^*(\eta_0)\}_{i=1}^n \overset{i.i.d.}{\sim} \text{Uniform}[0,1]$. We further rescale $\boldsymbol{\theta}^*(t)$ by setting $\theta_i^*(\eta_0) \leftarrow \frac{\psi^{-1}(0.9)}{\max_i \theta_i^*(\eta_0) - \min_i \theta_i^*(\eta_0)} \theta_i^*(\eta_0)$ and then set $\theta_i^*(\eta_0) \leftarrow \theta_i^*(\eta_0) - \text{avg}(\boldsymbol{\theta}^*(\eta_0))$. Here $\psi^{-1}(p) = \log(\frac{p}{1-p})$ is the inverse function of $\psi$. Recall that $P_{ij} = \psi(\theta_i - \theta_j)$ is the winning probability. So by rescaling $\boldsymbol{\theta}^*(t)$, we guarantee that at time $\eta_0$, the maximum winning probability is 0.9.

For each change point $\eta_k$, $k \geq 1$, we randomly sample a permutation $\pi : [n] \mapsto [n]$ from the collection of all $n$-permutations and set $\theta_i^*(\eta_k) = \theta_{\pi(i)}^*(\eta_{k-1})$ for $i \in [n]$. We consider the same settings for $(n, K, \Delta)$ with the same tuning parameters as in Section 4, and summarize our new simulation results in Table 3

| | $H(\hat{\eta}, \eta)$ | Time | $\hat{K} < K$ | $\hat{K} = K$ | $\hat{K} > K$ |
|---|---|---|---|---|---|
| **Setting (i)** | $n = 10, K = 3, \Delta = 500$, Random change | | | | |
| DPLR | 12.1 (13.3) | 62.4s (2.1) | 0 | 100 | 0 |
| WBS-GLR | 94.9 (174.8) | 33.6s (5.4) | 0 | 100 | 0 |
| **Setting (ii)** | $n = 20, K = 3, \Delta = 800$, Random change | | | | |
| DPLR | 23.9 (27.6) | 105.8s (4.2) | 0 | 100 | 0 |
| WBS-GLR | 251.7 (219.9) | 133.7s (14.7) | 0 | 40 | 60 |
| **Setting (iii)** | $n = 100, K = 2, \Delta = 1000$, Random change | | | | |
| DPLR | 43.1 (103.4) | 196.9s (3.9) | 1 | 99 | 0 |
| WBS-GLR | 133.0 (194.9) | 210.0s (16.6) | 0 | 76 | 24 |
| **Setting (iv)** | $n = 100, K = 3, \Delta = 2000$, Random change | | | | |
| DPLR | 28.3 (26.5) | 453.6s (9.2) | 0 | 100 | 0 |
| WBS-GLR | 459.4 (512.8) | 410.5s (48.7) | 0 | 53 | 47 |

Table 3: Comparison between DPLR and WBS-GLR under four different simulation settings with random signals. For the localization error and running time (in seconds), the averages over 100 trials are reported with standard errors in the brackets. The last three columns on the right record the number of trials in which $\hat{K} < K$, $\hat{K} = K$, and $\hat{K} > K$ respectively.

In what follows, we further investigate the effect of signal strength by restricting the random permutation at each change point to a subset of $[n]$, and analyze the performance of both methods while varying the size of permuted subsets. The results are summarized in Table 4, where 50% random permutation means at each change point $\eta_k$, only 50% of the entries of $\boldsymbol{\theta}^*(\eta_{k-1})$ are randomly selected and permuted to form $\boldsymbol{\theta}^*(\eta_k)$. Note that as the proportion of the randomly permuted entries increases, the random perturbation strength raises at the change points. As shown in Table 4, our algorithm DPLR is able to provide more accurate change point estimations as the random perturbation strength increases.

| Random permutation | Method | $H(\hat{\eta}, \eta)$ | Time | $\hat{K} < K$ | $\hat{K} = K$ | $\hat{K} > K$ |
|---|---|---|---|---|---|---|
| | | $n = 20, K = 3, \Delta = 800$ | | | | |
| 50% | DPLR | 362.8 (502.2) | 97.1s (10.4) | 27 | 67 | 6 |
| | WBS | 407.5 (336.8) | 137.2s (21.7) | 10 | 21 | 69 |
| 75% | DPLR | 114.4 (251.3) | 120.4s (4.4) | 8 | 91 | 1 |
| | WBS | 349.6 (261.8) | 141.8s (17.2) | 13 | 28 | 59 |
| 100% | DPLR | 23.9 (27.6) | 105.8s (4.2) | 0 | 100 | 0 |
| | WBS | 251.7 (219.9) | 133.7s (14.7) | 0 | 40 | 60 |

Table 4: Performance of DPLR and WBS-GLR under different signal strength. For the localization error and running time (in seconds), the average over 100 trials is shown with standard error in the bracket.

### A.3   Additional results for real data applications

#### A.3.1   Exploratory analysis

We start our analysis by fitting the BTL model on each season and drawing the path of fitted $\hat{\boldsymbol{\theta}}(\mathcal{I}_s)$, where $\mathcal{I}_s$ is the index interval for games in the $s$-th season in our range of interest, i.e., from season 1980-1981 to season 2015-2016. The resulting paths shown in Figure 1 are fairly noisy for interpretation and inference, and this is a strong evidence that the data is unstationary. In addition, these unstructured paths explain why we need some principled framework like change point models to analyze such unstationary data.

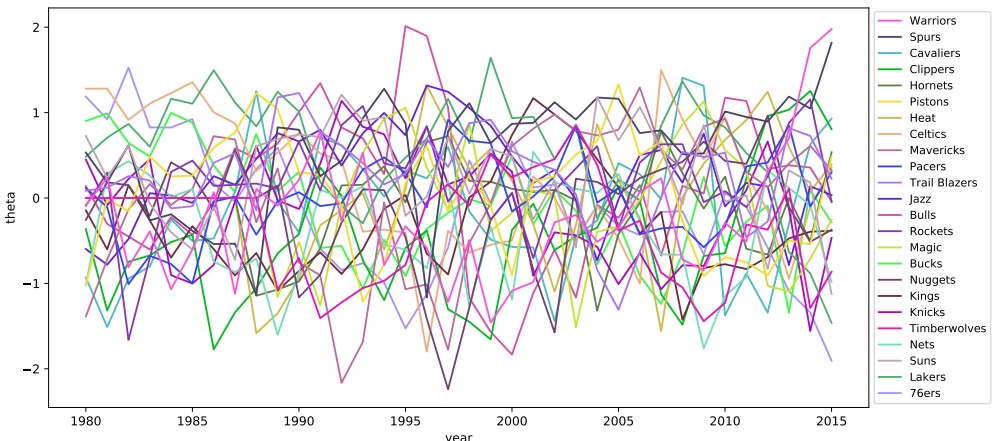

Figure 1: Path of $\hat{\boldsymbol{\theta}}(\mathcal{I}_s)$ for $\mathcal{I}_s$ being each season of the NBA data.

To get a rough sense of the number and locations of change points, we check the paths of the logarithm of generalized likelihood ratio statistics, which are shown in Figure 2. It should be noted that although the GLR paths suggest the existence and locations of two change points, we cannot rely on these observation. This is because when multiple change points exist, there will be cancellations effects and the GLR paths may not give consistent estimates of change points (Venkatraman, 1992). We can also see that splitting the data by odd and even indices does not affect the shape of the GLR path.

With all the information in the exploratory analysis, we apply our method DPLR to the dataset and summarize results in Section 5.

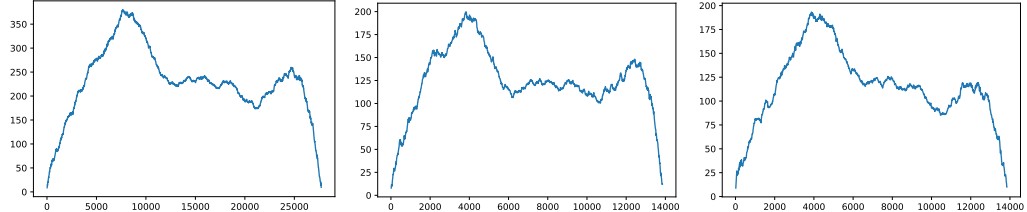

Figure 2: Path of (logarithmic) generalized loglikelihood ratio on NBA data. Left: GLR path on all samples; mid: GLR path on samples with odd indices; right: GLR path on samples with even indices.

#### A.3.2   Comparison with WBS-GLR

In this subsection, we apply the potential competitor, the likelihood-based WBS method (i.e. WBS-GLR), to the NBA data. For a fair comparison, we set the regularization tuning parameter $\gamma$ in the penalized logistic regression to be $0.1$, as we did in Section 5 for DPLR. However, as mentioned in Appendix A.1, WBS has another tuning parameter $M$, the number of random intervals to perform binary segmentation. So we apply WBS-GLR with $M \in \{50, 100, 150, 200, 250\}$, and the estimated change points with corresponding test errors (negative log-likelihoods) are summarized in Table 5.

Here, we use samples with odd time indices as training data and even time indices as test data. It can be seen from Table 5 that the choice of $M$ does not have a significant impact on change point estimation in this real data example. Therefore in what follows, we only discuss the results of WBS-GLR with $M = 200$.

| $M$ | Change point index | Change point season | Test errors |
|---|---|---|---|
| 50 | $[7728, 14628, 20700, 24564]$ | [S1990m, S1999, S2007, S2012] | 1796.9 |
| 100 | $[7728, 14628, 20700, 24564]$ | [S1990m, S1999, S2007, S2012] | 1796.9 |
| 150 | $[7728, 14628, 20700, 24564]$ | [S1990m, S1999, S2007, S2012] | 1796.9 |
| 200 | $[7728, 14352, 20700, 24564]$ | [S1990m, S1998m, S2007, S2012] | 1793.2 |
| 250 | $[7728, 14628, 20700, 24564]$ | [S1990m, S1999, S2007, S2012] | 1796.9 |

Table 5: The estimated change points with testing loss of WBS-GLR on the NBA data. S1980 means season 1980-1981, and S1990m means the middle of season 1990-1991

Then similar to Section 5, we fit a BTL model to each interval segmented by WBS-GLR, and summarize the results in Table 6. As we can see, WBS-GLR is able to detect several important change points in the NBA history, e.g., the dominance of Celtics and Lakers in 1980s, the Bulls dynasty in 1990s, and the rise of Spurs afterwards. However, compared with DPLR, WBS-GLR fails to detect the rise of Heat and Warriors. Therefore, the outcome of DPLR is more informative in this real application, which again confirms our findings in the simulation study in Section 4.

| S1980-S1990m | | S1990m-S1998m | | S1998m-S2006 | | S2007-S2011 | | S2012-S2015 | |
|---|---|---|---|---|---|---|---|---|---|
| Celtics | 1.1137 | Bulls | 0.9435 | Spurs | 0.904 | Lakers | 0.7579 | Spurs | 1.1659 |
| Lakers | 1.084 | Jazz | 0.7996 | Mavericks | 0.665 | Spurs | 0.701 | Clippers | 0.9448 |
| 76ers | 0.8049 | Suns | 0.5405 | Lakers | 0.5904 | Celtics | 0.6406 | Warriors | 0.9106 |
| Bucks | 0.7336 | Knicks | 0.5178 | Kings | 0.5103 | Magic | 0.6084 | Heat | 0.5149 |
| Pistons | 0.5074 | Rockets | 0.508 | Suns | 0.3677 | Mavericks | 0.605 | Rockets | 0.4703 |
| Trail Blazers | 0.4466 | Trail Blazers | 0.4931 | Timberwolves | 0.2767 | Nuggets | 0.458 | Mavericks | 0.3402 |
| Suns | 0.284 | Spurs | 0.4638 | Pistons | 0.2464 | Bulls | 0.2974 | Pacers | 0.3368 |
| Nuggets | 0.2294 | Cavaliers | 0.3415 | Jazz | 0.2266 | Suns | 0.28 | Trail Blazers | 0.2782 |
| Bulls | 0.1782 | Lakers | 0.3338 | Pacers | 0.1902 | Rockets | 0.2724 | Bulls | 0.2639 |
| Jazz | 0.1774 | Pacers | 0.241 | Rockets | 0.0024 | Jazz | 0.2499 | Nuggets | 0.0401 |
| Spurs | 0.1394 | Magic | 0.1824 | Trail Blazers | -0.0049 | Trail Blazers | 0.1843 | Jazz | -0.0495 |
| Rockets | 0.1252 | Hornets | 0.0923 | Heat | -0.0433 | Cavaliers | 0.1628 | Cavaliers | -0.0752 |
| Mavericks | 0.1004 | Heat | 0.0572 | 76ers | -0.0673 | Hornets | 0.0931 | Celtics | -0.1486 |
| Knicks | 0.0744 | Pistons | -0.1381 | Nets | -0.0807 | Heat | 0.081 | Hornets | -0.1522 |
| Warriors | -0.1406 | Warriors | -0.2101 | Hornets | -0.113 | 76ers | -0.157 | Nets | -0.2055 |
| Nets | -0.1751 | Celtics | -0.2326 | Bucks | -0.2183 | Pistons | -0.2651 | Knicks | -0.2865 |
| Pacers | -0.1857 | Nets | -0.3088 | Nuggets | -0.2676 | Warriors | -0.3028 | Suns | -0.296 |
| Cavaliers | -0.2179 | Bucks | -0.473 | Magic | -0.2993 | Pacers | -0.3475 | Bucks | -0.354 |
| Kings | -0.3197 | Clippers | -0.5024 | Knicks | -0.3218 | Bucks | -0.4778 | Pistons | -0.3591 |
| Clippers | -0.6276 | Kings | -0.5103 | Celtics | -0.3293 | Knicks | -0.6236 | Kings | -0.4707 |
| Timberwolves | -0.9485 | Nuggets | -0.6578 | Clippers | -0.4028 | Clippers | -0.6919 | Lakers | -0.5136 |
| Hornets | -1.0599 | Timberwolves | -0.6859 | Cavaliers | -0.4321 | Kings | -0.7288 | Timberwolves | -0.5649 |
| Magic | -1.1178 | 76ers | -0.7395 | Warriors | -0.5857 | Timberwolves | -0.8974 | Magic | -0.697 |
| Heat | -1.206 | Mavericks | -1.056 | Bulls | -0.8137 | Nets | -0.8998 | 76ers | -1.093 |

Table 6: Fitted $\hat{\theta}$ (rounded to the fourth decimal) for 24 selected teams in seasons 1980-2016 of the National Basketball Association. Teams are ranked by the MLE $\hat{\theta}$ on subsets splitted at the estimated change points given by the WBS-GLR method. S1980 means season 1980-1981, and S1990m means the middle of season 1990-1991.

### A.4   Other potential competitors

As we emphasized in Section 1 and Section 4, localizing potential change points in pairwise comparison data is an unsolved problem. Given the good performance of our proposed method DPLR in this paper, one might wonder if there exist other methods that perform well, or even better than DPLR, in some aspects. This section intends to present some of our explorations on two potential efficient methods, WBS-SST and WBS-Mean.

In what follows, we will demonstrate that both of them have crucial drawbacks. Specifically, WBS-Mean is not guaranteed to work for general comparison graphs, and works for general ranking models only under some constraints. WBS-SST works for general comparison graphs and ranking models, but requires relatively large sample size (i.e., $\Delta$) to work. Precise quantification of their performance can be an interesting direction for future works.

#### A.4.1   Based on the test statistic for SST class

Rastogi et al. (2020) consider the two sample testing problem for general pairwise comparison data. Suppose we observe pairwise comparison outcome matrices $\mathbf{X}$ and $\mathbf{Y}$ generated from two winning probability matrices $\mathbf{P}, \mathbf{Q} \in \mathbb{R}^{n \times n}$, respectively. They propose the following test statistic:

$$R_{SST} = \sum_{i=1}^{d} \sum_{j=1}^{d} \mathbb{I}_{ij} \frac{k_{ij}^q \left(k_{ij}^q - 1\right)\left(X_{ij}^2 - X_{ij}\right) + k_{ij}^p \left(k_{ij}^p - 1\right)\left(Y_{ij}^2 - Y_{ij}\right) - 2\left(k_{ij}^p - 1\right)\left(k_{ij}^q - 1\right)X_{ij}Y_{ij}}{\left(k_{ij}^p - 1\right)\left(k_{ij}^q - 1\right)\left(k_{ij}^p + k_{ij}^q\right)},$$
(A.3)

where $\mathbb{I}_{ij} = \mathbb{I}\left(k_{ij}^p > 1\right) \times \mathbb{I}\left(k_{ij}^q > 1\right)$, $k_{ij}^p = X_{ij} + X_{ji}$ and $k_{ij}^q = Y_{ij} + Y_{ji}$ are the number of comparisons between pairs.

We can use this test statistic to construct the loss $\mathcal{R}(t; s, e)$ in WBS (Algorithm 3), i.e.,

$$\mathcal{R}(t; s, e) = R_{SST}(\mathbf{X}([s, t)), \mathbf{Y}([t, e))),$$
(A.4)

and call this method WBS-SST (SST stands for strong stochastic transitive). When $\Delta$ is sufficiently large, WBS-SST performs fairly well with small computational cost, as is shown in Table 7.

**Issue with this approach.** However, When $\Delta$ is small, then many pairs in sampled intervals in WBS will have $k_{ij} \leq 1$, and the statistic would not be very powerful. See Figure 3 and Table 7.

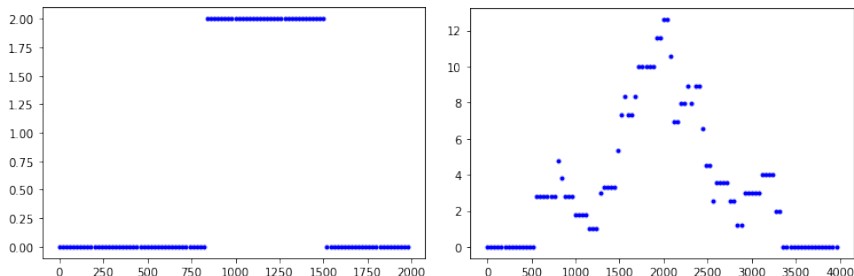

Figure 3: Loss path for WBS-SST when the sample size is not large enough. $n = 100$ with a single change point at the middle. $\Delta = 1000$ (left), $\Delta = 2000$ (right).

To see that reason, notice that

$$\mathbb{E}[R_{SST}|\mathbf{k}^p, \mathbf{k}^q] = \sum_{i=1}^{d} \sum_{j=1}^{d} \mathbb{I}_{ij} \frac{k_{ij}^q \left(k_{ij}^q - 1\right) k_{ij}^p \left(k_{ij}^p - 1\right)\left(P_{ij}^2 + Q_{ij}^2 - 2P_{ij}Q_{ij}\right)}{\left(k_{ij}^p - 1\right)\left(k_{ij}^q - 1\right)\left(k_{ij}^p + k_{ij}^q\right)}$$
$$= \sum_{i=1}^{d} \sum_{j=1}^{d} \mathbb{I}_{ij} \frac{k_{ij}^q k_{ij}^p}{k_{ij}^p + k_{ij}^q}\left(P_{ij} - Q_{ij}\right)^2.$$
(A.5)

When the comparison graph is a complete graph and compared pairs $\{(i_t, j_t)\}_{t \in [T]}$ are i.i.d. samples from the edge set $E_{full} := \{(i, j) : 1 \leq i < j \leq n\}$, the expectation of $R$ is (without the loss of

generality, assume that $(1,2) \in E_{full}$)

$$\mathbb{E}[R_{SST}] = \|P - Q\|_F^2 \mathbb{E}[\mathbb{I}_{1,2} \frac{k_{1,2}^q k_{1,2}^p}{k_{1,2}^p + k_{1,2}^q}]. \tag{A.6}$$

The two equations above illustrate why WBS-SST does nor perform well in small-SNR cases.

### A.4.2   Based on the Borda count

Borda count is a popular method in practice for ranking, due to its efficiency and generality (Shah and Wainwright, 2018). Given an interval $\mathcal{I}$, the normalized Borda count vector is defined as

$$\beta(\mathcal{I})_i = \frac{1}{|\mathcal{I}|}[N_w(i; \mathcal{I}) - N_l(i; \mathcal{I})], \forall i \in [n], \tag{A.7}$$

where $N_w(i; \mathcal{I})$ and $N_l(i; \mathcal{I})$ are the number of wining and loss of item $i$ in comparisons over the interval $\mathcal{I}$.

Since it is well-known in ranking literature that Borda count is not guaranteed to give consistent ranking for general comparison graphs, we only consider the complete graph here. When the comparison graph is a complete graph and compared pairs are i.i.d. samples from the edge set, and there is no change point in $\mathcal{I}$, the expectation of $\beta(\mathcal{I})_i$ is

$$\mathbb{E}[\beta(\mathcal{I})_i] = \frac{2}{n(n-1)} \sum_{j \neq i} (P_{ij} - P_{ji}) = \frac{2}{n(n-1)} \sum_{j \neq i} (2P_{ij} - 1), \tag{A.8}$$

where $P_{ij} = \mathbb{P}[i \text{ beats } j]$.

If we treat $\beta(\mathcal{I})$ as a sample mean of a random variable, we can construct the CUSUM statistic at $t \in \mathcal{I} = [s, e)$ as

$$\mathcal{R}_{Borda}(t; [s, e)) = \frac{(t-s)(e-t)}{e-s} \|\boldsymbol{\beta}([s,t)) - \boldsymbol{\beta}([t,e))\|_2^2. \tag{A.9}$$

To compared this statistic with Equation (A.3), we assume there is a single change point $\eta \in [s, e)$ and check the statistic at $\eta$. By Equation (A.8), the population version of the statistic is

$$\begin{aligned}
\widetilde{\mathcal{R}}_{Borda}(\eta; [s, e)) &= \frac{(\eta-s)(e-\eta)}{e-s} \|\mathbb{E}\boldsymbol{\beta}([s,\eta)) - \mathbb{E}\boldsymbol{\beta}([\eta,e))\|_2^2 \\
&= \frac{(\eta-s)(e-\eta)}{e-s} \cdot \frac{2}{n(n-1)} \sum_{i \in [n]} [\sum_{j \neq i} (P_{ij} - Q_{ij})]^2
\end{aligned} \tag{A.10}$$

where $\mathbf{P}, \mathbf{Q}$ are the winning probability matrices before and after the change point $\eta$.

**Issue with this approach.**   With Equation (A.10), we can construct examples such that the population version of the CUSUM statistic is very small or even zero at the true change point $\eta$. For instance, let $n = 3$ and

$$\mathbf{P} = \begin{bmatrix} 0.5 & 0.6 & 0.8 \\ 0.4 & 0.5 & 0.7 \\ 0.2 & 0.3 & 0.5 \end{bmatrix}, \quad \mathbf{Q} = \begin{bmatrix} 0.5 & 0.55 & 0.85 \\ 0.45 & 0.5 & 0.65 \\ 0.15 & 0.35 & 0.5 \end{bmatrix}, \tag{A.11}$$

then both $\mathbf{P}, \mathbf{Q}$ are strong-stochastic-transitive matrices (see Shah and Wainwright (2018) for details) and the population CUSUM $\widetilde{\mathcal{R}}_{Borda}(\eta; [s, e)) = 0$ at $\eta$. Figure 4 compares paths of the loss for WBS-Mean and WBS-SST under the choice of $\mathbf{P}, \mathbf{Q}$ above, where there is a single change point at $\eta = 1000$.

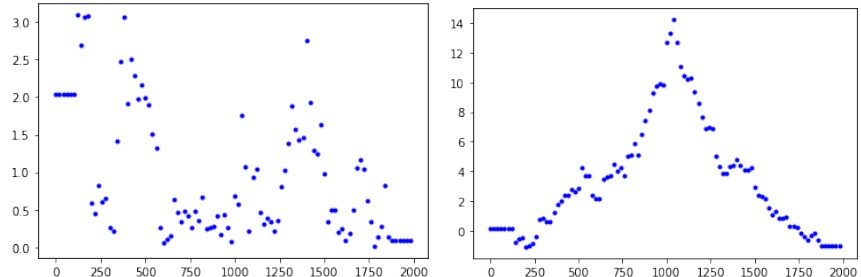

Figure 4: Loss path for WBS-Mean (left) and WBS-SST (right).

### A.4.3   Numerical performance

Table 7 compares the performance of WBS-SST and WBS-Mean with the two methods presented in the main text, under the identical setting in Section 4. The setting is sketched below for convenience.

| | $H(\hat{\eta}, \eta)$ | Time | $\hat{K} < K$ | $\hat{K} = K$ | $\hat{K} > K$ |
|---|---|---|---|---|---|
| **Setting (i)** | $n = 10, K = 3, \Delta = 500$, Change (I, II, III) | | | | |
| DPLR | 9.2 (9.1) | 49.7s (0.7) | 0 | 100 | 0 |
| WBS-Mean | 15.4 (8.4) | 0.2s (0.05) | 0 | 100 | 0 |
| WBS-SST | 16.2 (11.4) | 0.4s (0.2) | 0 | 100 | 0 |
| WBS-GLR | 15.2 (7.9) | 31.9s (3.9) | 0 | 100 | 0 |
| **Setting (ii)** | $n = 20, K = 3, \Delta = 800$, Change (I, II, III) | | | | |
| DPLR | 9.0 (9.9) | 118.5s (2.2) | 0 | 100 | 0 |
| WBS-Mean | 5.8 (11.4) | 0.5s (0.1) | 0 | 100 | 0 |
| WBS-SST | 19.4 (22.3) | 1.7s (0.5) | 0 | 100 | 0 |
| WBS-GLR | 240.5 (220.3) | 144.2s (12.5) | 0 | 40 | 60 |
| **Setting (iii)** | $n = 100, K = 2, \Delta = 1000$, Change (I, II) | | | | |
| DPLR | 13.4 (14.4) | 167.4s (3.3) | 0 | 100 | 0 |
| WBS-Mean | 22.9 (98.4) | 0.6s (0.04) | 1 | 99 | 0 |
| WBS-SST | $\infty$ (NA) | 3.9s (0.4) | 100 | 0 | 0 |
| WBS-GLR | 111.9 (195.6) | 215.9s (17.0) | 0 | 79 | 21 |
| **Setting (iv)** | $n = 100, K = 3, \Delta = 2000$, Change (I, II, III) | | | | |
| DPLR | 12.4 (12.1) | 402.4s (7.4) | 0 | 100 | 0 |
| WBS-Mean | 17.9 (6.1) | 0.9s (0.06) | 0 | 100 | 0 |
| WBS-SST | 1116.3 (694.8) | 19.3s (1.9) | 57 | 42 | 1 |
| WBS-GLR | 412.3 (495.5) | 400.0s (40.9) | 0 | 57 | 43 |

Table 7: Comparison of DPLR and three WBS-based methods under four different simulation settings. 100 trials are conducted in each setting. For the localization error and running time (in seconds), the average over 100 trials is shown with standard error in the bracket. The three columns on the right record the number of trials in which $\hat{K} < K$, $\hat{K} = K$, and $\hat{K} > K$ respectively.

**Simulation settings.**   For $1 < i \le n$, we set $\theta_i^*(\eta_0) = \theta_1^*(\eta_0) + (i - 1)\delta$ with some constant $\delta$. In each experiment, we set $\delta$ first and then set $\theta_1^*(\eta_0)$ to make $\mathbf{1}_n^\top \boldsymbol{\theta}^*(\eta_0) = 0$. The value of $\delta$ guarantees that the maximum winning probability is 0.9. We consider three types of changes:

Type I (reverse): $\theta_i^*(\eta_k) = \theta_{n+1-i}^*(\eta_0)$.

Type II (block-reverse): $\theta_i^*(\eta_k) = \theta_{[\frac{n}{2}]+1-i}^*(\eta_0)$ for $i \le [\frac{n}{2}]$; $\theta_i^*(\eta_k) = \theta_{[\frac{n}{2}]+n+1-i}^*(\eta_0)$ for $i > [\frac{n}{2}]$.

Type III (block exchange): $\theta_i^*(\eta_k) = \theta_{i+[\frac{n}{2}]}^*(\eta_0)$ for $i \le [\frac{n}{2}]$; $\theta_i^*(\eta_k) = \theta_{i-[\frac{n}{2}]}^*(\eta_0)$ for $i > [\frac{n}{2}]$.

We consider four simulation settings. For each setting, we have $T = (K + 1)\Delta$ and the change points locate at $\eta_i = i\Delta$ for $i \in [K]$. To describe the true parameter at each change point, we use an ordered tuple. For instance, (I, II, III, I) means that $K = 4$ and the true parameters at $\eta_1, \eta_2, \eta_3, \eta_4$ are determined based on $\boldsymbol{\theta}^*(\eta_0)$ and the change type I, II, III, and I, respectively.

# B   Appendix: Proof

This section has three parts:

1. Appendix B.1 contains the proof of two theorems in Section 3.
2. Appendix B.2 contains propositions used throughout the paper with proof.
3. Appendix B.3 contains all technical lemmas with proof.

## B.1   Proof of main theorems

*Proof of Theorem 3.2.* The theorem is a straightforward conclusion of Proposition B.2 and Proposition B.1. More specifically, conclusion 3 and 4 of Proposition B.2 guarantee that $K \le |\hat{\mathcal{P}}| \le 3K$ with probability at least $1 - (Tn)^{-2}$ and Proposition B.1 further confirms the consistency of $\hat{K}$. Then conclusion 1 and 2 of Proposition B.2 control the localization error. □

*Proof of Theorem 3.3.* The theorem is a straightforward conclusion of Theorem 3.2 that quantifies the localization error of outputs of dynamic programming and Proposition B.3 that shows the improvement of local refinement. □

## B.2   Main propositions

**Proposition B.1** (Consistency of $\hat{K}$). *Let $\hat{\mathcal{P}}$ be the estimator of change points in Equation* (3.1). *Assume $K \le |\hat{\mathcal{P}}| \le 3K$. Under all assumptions above, it holds with probability at least $1 - (Tn)^{-2}$ that $|\hat{\mathcal{P}}| = K$.*

*Proof.* For a sequence of strictly increasing integer time points $\{\eta_j'\}_{j \in [J+1]}$ with $\eta_0' = 1$ and $\eta_{J+1}' = T + 1$, let $\mathcal{I}_j = [\eta_{j-1}', \eta_j')$ and

$$L(\{\eta_j'\}_{j \in [J+1]}) = \sum_{j \in [J+1]} L(\hat{\boldsymbol{\theta}}(\mathcal{I}_j), \mathcal{I}_j),$$

where $\hat{\boldsymbol{\theta}}(\mathcal{I}_j) := \arg\min_{\boldsymbol{\theta} \in \Theta_B} L(\boldsymbol{\theta}, \mathcal{I}_j)$. Furthermore, when $\{\eta_k\}_{k \in [K]} \subset \{\eta_j'\}_{j \in [J+1]}$ so that $\theta^*(t)$ remains unchanged in each interval $\mathcal{I}_j$, we can define the risk of true parameters

$$L^*(\{\eta_j'\}_{j \in [J+1]}) = \sum_{j \in [J+1]} L(\boldsymbol{\theta}^*(\mathcal{I}_j), \mathcal{I}_j).$$

Let $\{\hat{\eta}_k\}_{k \in [\hat{K}]}$ be the change points given by the estimator $\hat{\mathcal{P}}$ and $\mathbf{Sort}(\cdot)$ be an operator on finite ordered tuple of scalars such that $\mathbf{Sort}((a_1, \ldots, a_m)) = (a_{(1)}, \ldots, a_{(m)})$ where $a_{(i)} \le a_{(j)}$ for $i < j$ and $\{a_{(i)}\}_{i \in [m]} = \{a_i\}_{i \in [m]}$. Then a sufficient condition for $|\hat{\mathcal{P}}| = K$ is

$$
\begin{aligned}
&L^*(\eta_1, \cdots, \eta_K) + K\gamma \\
\ge\ &L(\eta_1, \cdots, \eta_K) + K\gamma && \text{(B.1)} \\
\ge\ &L(\hat{\eta}_1, \cdots, \hat{\eta}_{\hat{K}}) + \hat{K}\gamma && \text{(B.2)} \\
\ge\ &L^*(\mathbf{Sort}(\hat{\eta}_1, \cdots, \hat{\eta}_{\hat{K}}, \eta_1, \cdots, \eta_K)) + \hat{K}\gamma - CKp_{lb}^{-2}\frac{nd_{\max}}{\lambda_2(L_\mathcal{G})}\log(Tn), && \text{(B.3)}
\end{aligned}
$$

and

$$L^*(\eta_1, \cdots, \eta_K) \le L^*(\mathbf{Sort}(\hat{\eta}_1, \cdots, \hat{\eta}_{\hat{K}}, \eta_1, \cdots, \eta_K)) + C_1 Kp_{lb}^{-2}\frac{nd_{\max}}{\lambda_2(L_\mathcal{G})}\log(Tn). \quad \text{(B.4)}$$

In fact, if $\hat{K} \ge K + 1$, under the conditions above and the assumption that $|\hat{\mathcal{P}}| \le 3K$, we have

$$\gamma \le (\hat{K} - K)\gamma \le C_2 Kp_{lb}^{-2}\frac{nd_{\max}}{\lambda_2(L_\mathcal{G})}\log(Tn),$$

which is contradictory to the definition $\gamma = C_\gamma(K+1)p_{lb}^{-2}\frac{nd_{\max}}{\lambda_2(L_\mathcal{G})}\log(Tn)$ for sufficiently large $C_\gamma$.

Now we prove that the sufficient condition holds with probability at least $1 - (Tn)^{-2}$. Equation (B.1) is a straightforward conclusion of the definition $\hat{\boldsymbol{\theta}}(\mathcal{I}_j) := \arg\min_{\boldsymbol{\theta}\in\Theta_B} L(\boldsymbol{\theta}, \mathcal{I}_j)$ and Equation (B.2) is implied by the definition of $\hat{\mathcal{P}}$ in Equation (3.1).

Equation (B.4) is guaranteed by Lemma B.9 because for any interval $\mathcal{I}$ determined by endpoints that are two consecutive points in $\mathbf{Sort}(\hat{\eta}_1, \cdots, \hat{\eta}_{\hat{K}}, \eta_1, \cdots, \eta_K)$, there will not be any true change point in the interior of $\mathcal{I}$.

For Equation (B.3), notice that by Proposition B.2, with probability $1 - (Tn)^{-4}$, there are at most two change points in $\mathcal{I}$. Therefore, Lemma B.7 ensures that

$$L(\hat{\eta}_1, \cdots, \hat{\eta}_{\hat{K}}) \geq L^*(\mathbf{Sort}(\hat{\eta}_1, \cdots, \hat{\eta}_{\hat{K}}, \eta_1, \cdots, \eta_K)) - CKp_{lb}^{-2}\frac{nd_{\max}}{\lambda_2(L_{\mathcal{G}})}\log(Tn).$$

$\square$

**Proposition B.2** (Four cases). *Let $\hat{\mathcal{P}}$ be the estimator of change points in Equation (3.1). Under Assumption 3.1 and Assumption B.14, with probability at least $1 - (Tn)^{-2}$ the following four events hold uniformly for all $\mathcal{I} = (s, e) \in \hat{\mathcal{P}}$:*

1. *If $\mathcal{I}$ contains only one change point $\eta$, then for some universal constant $C > 0$,*

$$\min\{\eta - s, e - \eta\} \leq Cp_{lb}^{-2}\frac{|E|}{\lambda_2(L_{\mathcal{G}})}[\gamma + \frac{nd_{\max}}{\lambda_2(L_{\mathcal{G}})}\log(Tn)].$$

2. *If $\mathcal{I}$ contains exactly two change points $\eta_k$ and $\eta_{k+1}$, then for some universal constant $C > 0$,*

$$\min\{\eta_k - s, e - \eta_{k+1}\} \leq Cp_{lb}^{-2}\frac{|E|}{\lambda_2(L_{\mathcal{G}})}[\gamma + \frac{nd_{\max}}{\lambda_2(L_{\mathcal{G}})}\log(Tn)].$$

3. *If $|\hat{\mathcal{P}}| > 1$, then for any two consecutive intervals $\mathcal{I}$ and $\mathcal{J}$ in $\hat{\mathcal{P}}$, the joint interval $\mathcal{I} \cup \mathcal{J}$ contains at least one change point.*

4. *Interval $\mathcal{I}$ does not contain more than two change points.*

*Proof.* Conclusion 1 is implied by Lemma B.11 and conclusion 2 is guaranteed by Lemma B.12. Conclusion 4 is a direct consequence of Lemma B.13 and the definition of $\hat{\mathcal{P}}$.

To prove conclusion 3, assume instead that there is no true change point in $\mathcal{I} \cup \mathcal{J}$. Then by Lemma B.10 we have

$$L(\hat{\boldsymbol{\theta}}(\mathcal{I}), \mathcal{I}) + L(\hat{\boldsymbol{\theta}}(\mathcal{J}), \mathcal{J}) + \gamma \geq L(\boldsymbol{\theta}^*(\mathcal{I} \cup \mathcal{J}), \mathcal{I} \cup \mathcal{J}) \geq L(\hat{\boldsymbol{\theta}}(\mathcal{I} \cup \mathcal{J}), \mathcal{I} \cup \mathcal{J}),$$

which is contradictory to the definition of $\hat{\mathcal{P}}$. $\square$

**Proposition B.3** (Local refinement). *Consider the local refinement procedure given in Algorithm 2, that is,*

$$\left(\hat{\eta}_k, \hat{\boldsymbol{\theta}}^{(1)}, \hat{\boldsymbol{\theta}}^{(2)}\right) = \argmin_{\substack{\eta\in\{s_k+1,\ldots,e_k-1\} \\ \boldsymbol{\theta}^{(1)},\boldsymbol{\theta}^{(2)}\in\Theta_B}} \left\{ \sum_{t=s_k+1}^{\eta} \ell_t(\boldsymbol{\theta}^{(1)}) + \sum_{t=\eta+1}^{e_k} \ell_t(\boldsymbol{\theta}^{(2)}) \right\}, \qquad \text{(B.5)}$$

*where $s_k = 2\widetilde{\eta}_{k-1}/3 + \widetilde{\eta}_k/3$ and $e_k = \widetilde{\eta}_k/3 + 2\widetilde{\eta}_{k+1}/3$ and $\ell_t(\boldsymbol{\theta})$ is the negative log-likelihood given in Equation (2.6). Suppose the input $\{\widetilde{\eta}_k\}_{k\in[\tilde{K}]}$ satisfies $\tilde{K} = K$ and*

$$\max_{k\in[K]} |\widetilde{\eta}_k - \eta_k| \leq \Delta/5.$$

*Let $\{\hat{\eta}_k\}_{k\in[K]}$ be the output. Then it holds with probability at least $1 - (Tn)^{-2}$ that*

$$\max_{k\in[K]} |\hat{\eta}_k - \eta_k| \leq C\frac{|E|nd_{\max}}{p_{lb}^2\kappa^2\lambda_2^2(L_{\mathcal{G}})}\log(Tn). \qquad \text{(B.6)}$$

*Proof.* For each $k \in [K]$, let $\hat{\boldsymbol{\theta}}(t) = \hat{\boldsymbol{\theta}}^{(1)}$ if $s_k < t \leq \hat{\eta}_k$ and $\hat{\boldsymbol{\theta}}(t) = \hat{\boldsymbol{\theta}}^{(2)}$ otherwise, and $\boldsymbol{\theta}^*(t)$ be the true parameter at time point $t$. First we show that under conditions $\tilde{K} = K$ and $\max_{k \in [K]} |\tilde{\eta}_k - \eta_k| \leq \Delta/5$, there is only one true change point $\eta_k$ in $(s_k, e_k)$. It suffices to show that

$$|\tilde{\eta}_k - \eta_k| \leq \frac{2}{3}(\tilde{\eta}_{k+1} - \tilde{\eta}_k), \text{ and } |\tilde{\eta}_{k+1} - \eta_{k+1}| \leq \frac{1}{3}(\tilde{\eta}_{k+1} - \tilde{\eta}_k). \tag{B.7}$$

Denote $R = \max_{k \in [K]} |\tilde{\eta}_k - \eta_k|$, then

$$\begin{aligned}
\tilde{\eta}_{k+1} - \tilde{\eta}_k &= \tilde{\eta}_{k+1} - \eta_{k+1} + \eta_{k+1} - \eta_k + \eta_k - \tilde{\eta}_k \\
&= (\eta_{k+1} - \eta_k) + (\tilde{\eta}_{k+1} - \eta_{k+1}) + (\eta_k - \tilde{\eta}_k) \in [\eta_{k+1} - \eta_k - 2R, \eta_{k+1} - \eta_k + 2R].
\end{aligned}$$

Therefore, Equation (B.7) is guaranteed as long as

$$R \leq \frac{1}{3}(\Delta - 2R),$$

which is equivalent to $R \leq \Delta/5$.

Now without loss of generality, assume that $s_k < \eta_k < \hat{\eta}_k < e_k$. Denote $\mathcal{I}_k = \{s_k + 1, \cdots, e_k\}$. Consider two cases:

**Case 1** If

$$\hat{\eta}_k - \eta_k < \max\{Cp_{lb}^{-2} \frac{|E|nd_{\max}}{\lambda_2^2(L_{\mathcal{G}})} \log(Tn), Cp_{lb}^{-2} \log(Tn)/\kappa^2\},$$

then the proof is done.

**Case 2** If

$$\hat{\eta}_k - \eta_k \geq \max\{Cp_{lb}^{-2} \frac{|E|nd_{\max}}{\lambda_2^2(L_{\mathcal{G}})} \log(Tn), Cp_{lb}^{-2} \log(Tn)/\kappa^2\},$$

then we proceed to prove that $|\hat{\eta}_k - \eta_k| \leq C_1 \frac{|E|nd_{\max}}{p_{lb}^2 \kappa^2 \lambda_2^2(L_{\mathcal{G}})} \log(Tn)$ with probability at least $1 - (Tn)^{-3}$. Then we either prove the result or get an contradiction, and complete the proof in either case.

By the definition of $\hat{\eta}_k, \hat{\boldsymbol{\theta}}^{(1)}$, and $\hat{\boldsymbol{\theta}}^{(2)}$, we have

$$\sum_{t \in \mathcal{I}_k} \ell_t(\hat{\boldsymbol{\theta}}(t)) \leq \sum_{t \in \mathcal{I}_k} \ell_t(\boldsymbol{\theta}^*(t)).$$

By Lemma B.8, this implies that

$$ce^{-2B} \sum_{t \in \mathcal{I}_k} [\mathbf{x}(t)^\top \Delta(t)]^2 \leq \sum_{t \in \mathcal{I}_k} \epsilon_t \mathbf{x}(t)^\top \Delta(t), \tag{B.8}$$

where $\Delta(t) := \hat{\boldsymbol{\theta}}(t) - \boldsymbol{\theta}^*(t)$ and $\epsilon_t := y_t - \frac{\exp(\mathbf{x}(t)^\top \boldsymbol{\theta}^*(t))}{1 + \exp(\mathbf{x}(t)^\top \boldsymbol{\theta}^*(t))}$. For the cross term, by Lemma B.19 we have

$$\begin{aligned}
\sum_{t \in \mathcal{I}_k} \epsilon_t \mathbf{x}(t)^\top \Delta(t) &= \sum_{i \in [n]} \left\{ \left| \frac{\sum_{t \in \mathcal{I}_k} \epsilon_t x_i(t) \Delta_i(t)}{\sqrt{\sum_{t \in \mathcal{I}_k} \Delta_i(t)^2}} \right| \sqrt{\sum_{t \in \mathcal{I}_k} \Delta_i(t)^2} \right\} \\
&\leq \sup_{i \in [n]} \left| \frac{\sum_{t \in \mathcal{I}_k} \epsilon_t x_i(t) \Delta_i(t)}{\sqrt{\sum_{t \in \mathcal{I}_k} \Delta_i(t)^2}} \right| \sum_{i \in [n]} \sqrt{\sum_{t \in \mathcal{I}_k} \Delta_i(t)^2} \\
&\leq C\sqrt{\frac{d_{\max}}{|E|} \log(Tn)} \sum_{i \in [n]} \sqrt{\sum_{t \in \mathcal{I}_k} (\hat{\theta}_i - \theta_i^*(t))^2} \\
&\leq C\sqrt{\frac{nd_{\max}}{|E|} \log(Tn)} \sqrt{\sum_{t \in \mathcal{I}_k} \|\Delta(t)\|_2^2}. \tag{B.9}
\end{aligned}$$

Equation (B.8) and Equation (B.9) together imply that

$$ce^{-2B} \sum_{t \in \mathcal{I}_k} [\mathbf{x}(t)^\top \Delta(t)]^2 \leq C\sqrt{\frac{nd_{\max}}{|E|} \log(Tn)} \sqrt{\sum_{t \in \mathcal{I}_k} \|\Delta(t)\|_2^2}. \tag{B.10}$$

Let
$$\mathcal{J}_1 = (s_k, \eta_k], \ \mathcal{J}_2 = (\eta_k, \hat{\eta}_k], \ \mathcal{J}_3 = (\hat{\eta}_k, e_k].$$
Under Assumption 3.1 and the condition of the proposition, it holds that $\min\{|\mathcal{J}_1|, |\mathcal{J}_3|\} \geq C_0 \frac{|E| \log(Tn)}{\lambda_2(L_{\mathcal{G}})}$. Thus, by Lemma B.17, with probability at leat $1 - (Tn)^{-3}$, we have
$$\sum_{t \in \mathcal{I}_k} [\mathbf{x}(t)^\top \Delta(t)]^2 \geq \frac{c_1 \lambda_2(L_{\mathcal{G}})}{|E|} \sum_{t \in \mathcal{I}_k} \|\Delta(t)\|_2^2.$$
The inequality above leads to
$$\sum_{t \in \mathcal{I}_k} \|\hat{\boldsymbol{\theta}}(t) - \boldsymbol{\theta}^*(t)\|_2^2 \leq C p_{lb}^{-2} \frac{|E| n d_{\max}}{\lambda_2^2(L_{\mathcal{G}})} \log(Tn).$$
Recall that we defined $\boldsymbol{\theta}^{(1)} = \boldsymbol{\theta}^*(\eta_k - 1)$ and $\boldsymbol{\theta}^{(2)} = \boldsymbol{\theta}^*(\eta_k)$. Then we have
$$\sum_{t \in \mathcal{I}_k} \|\hat{\boldsymbol{\theta}}(t) - \boldsymbol{\theta}^*(t)\|_2^2 = |\mathcal{J}_1| \|\hat{\boldsymbol{\theta}}^{(1)} - \boldsymbol{\theta}^{(1)}\|_2^2 + |\mathcal{J}_2| \|\hat{\boldsymbol{\theta}}^{(1)} - \boldsymbol{\theta}^{(2)}\|_2^2 + |\mathcal{J}_3| \|\hat{\boldsymbol{\theta}}^{(2)} - \boldsymbol{\theta}^{(2)}\|_2^2.$$
Since $|\mathcal{J}_1| = \eta_k - s_k \geq c_0 \Delta$ with some constant $c_0$ under Assumption 3.1, we have
$$\Delta \|\hat{\boldsymbol{\theta}}^{(1)} - \boldsymbol{\theta}^{(1)}\|_2^2 \leq c_0 |\mathcal{J}_1| \|\hat{\boldsymbol{\theta}}^{(1)} - \boldsymbol{\theta}^{(1)}\|_2^2 \leq c_1 p_{lb}^{-2} \frac{|E| n d_{\max}}{\lambda_2^2(L_{\mathcal{G}})} \log(Tn) \leq c_2 \Delta \kappa^2, \quad \text{(B.11)}$$
with some constant $c_2 \in (0, 1/4)$, where the last inequality is due to the fact that $\mathcal{B}_T \to \infty$. Thus we have
$$\|\hat{\boldsymbol{\theta}}^{(1)} - \boldsymbol{\theta}^{(1)}\|_2^2 \leq c_2 \kappa^2.$$
Triangle inequality gives
$$\|\hat{\boldsymbol{\theta}}^{(1)} - \boldsymbol{\theta}^{(2)}\|_2 \geq \|\boldsymbol{\theta}^{(1)} - \boldsymbol{\theta}^{(2)}\|_2 - \|\hat{\boldsymbol{\theta}}^{(1)} - \boldsymbol{\theta}^{(1)}\|_2 \geq \kappa/2.$$
Therefore, $\kappa^2 |\mathcal{J}_2|/4 \leq |\mathcal{J}_2| \|\hat{\boldsymbol{\theta}}^{(1)} - \boldsymbol{\theta}^{(2)}\|_2^2 \leq C p_{lb}^{-2} \frac{|E| n d_{\max}}{\lambda_2^2(L_{\mathcal{G}})} \log(Tn)$ and
$$|\hat{\eta}_k - \eta_k| = |\mathcal{J}_2| \leq \frac{C p_{lb}^{-2} |E| n d_{\max} \log(Tn)}{\lambda_2^2(L_{\mathcal{G}}) \kappa^2}.$$

$\square$

**Proposition B.4.** *Let* $\mathbf{P}(\boldsymbol{\theta})$ *be the winning probability matrix induced by* $\boldsymbol{\theta}$. *For* $\boldsymbol{\theta}^{(1)}, \boldsymbol{\theta}^{(2)} \in \Theta_B$, *it holds that*
$$\frac{n p_{lb}^2}{16} \|\boldsymbol{\theta}^{(1)} - \boldsymbol{\theta}^{(2)}\|_2^2 \leq \|\mathbf{P}(\boldsymbol{\theta}^{(1)}) - \mathbf{P}(\boldsymbol{\theta}^{(2)})\|_F^2 \leq \frac{n}{16} \|\boldsymbol{\theta}^{(1)} - \boldsymbol{\theta}^{(2)}\|_2^2, \quad \text{(B.12)}$$
*where* $p_{lb} = \frac{e^{-2B}}{1 + e^{-2B}}$.

*Proof.* This result has been shown in Shah and Wainwright (2018) (In the proof of Theorem 4). We include it here for completeness.

Denote $\psi(t) = \frac{1}{1 + e^{-t}}$. For any pair $(i, j) \in [n]^2$, by the mean value theorem we have
$$|P_{ij}(\boldsymbol{\theta}^{(1)}) - P_{ij}(\boldsymbol{\theta}^{(2)})| = |\psi(\theta_i^{(1)} - \theta_j^{(1)}) - \psi(\theta_i^{(2)} - \theta_j^{(2)})|$$
$$= |\psi'(\xi)| |(\theta_i^{(1)} - \theta_j^{(1)}) - (\theta_i^{(2)} - \theta_j^{(2)})|,$$
where $\xi$ is a scalar between $(\theta_i^{(1)} - \theta_j^{(1)})$ and $(\theta_i^{(2)} - \theta_j^{(2)})$. Since $\psi'(t) = \psi(t)(1 - \psi(t)) \in (\frac{1}{4e^{2B}}, \frac{1}{4}]$ for $t \in [-2B, 2B]$, we have
$$\frac{1}{4e^{2B}} |(\theta_i^{(1)} - \theta_j^{(1)}) - (\theta_i^{(2)} - \theta_j^{(2)})| \leq |P_{ij}(\boldsymbol{\theta}^{(1)}) - P_{ij}(\boldsymbol{\theta}^{(2)})| \leq \frac{1}{4} |(\theta_i^{(1)} - \theta_j^{(1)}) - (\theta_i^{(2)} - \theta_j^{(2)})|.$$
By the property of Graph Laplacian and the fact that $\mathbf{1}_n^\top \boldsymbol{\theta}^{(i)} = 0$, $i = 1, 2$, we have
$$\sum_{i, j \in [n]^2} [(\theta_i^{(1)} - \theta_j^{(1)}) - (\theta_i^{(2)} - \theta_j^{(2)})]^2 = (\boldsymbol{\theta}^{(1)} - \boldsymbol{\theta}^{(2)})^\top [n \mathbf{I}_n - \mathbf{1}_n \mathbf{1}_n^\top](\boldsymbol{\theta}^{(1)} - \boldsymbol{\theta}^{(2)}) \quad \text{(B.13)}$$
$$= n \|\boldsymbol{\theta}^{(1)} - \boldsymbol{\theta}^{(2)}\|_2^2. \quad \text{(B.14)}$$
Combining the results above gives the conclusion.                          $\square$

**Proposition B.5** (Single change point). *Suppose we observe $\{(\mathbf{x}(t), y_t)\}_{t \in [T]}$ following model (2.1) and (2.5) and there is a single change point $\eta \in (1, T)$. In addition, assume that*

$$\Delta := \min\{\eta - 1, T - \eta\} \geq \mathcal{B}_T \frac{|E| n d_{\max}}{p_{lb}^2 \kappa^2 \lambda_2^2(L_{\mathcal{G}})} \log(Tn), \tag{B.15}$$

*for a diverging sequence $\{\mathcal{B}_T\}_{T \in \mathbb{Z}_+}$. Let the estimator $\hat{\eta}$ of the change point be*

$$\hat{\eta} = \arg\min_{\eta \in [T]} \left\{ \min_{\boldsymbol{\theta}^{(1)} \in \Theta_B} \sum_{t=1}^{\eta} \ell_t(\boldsymbol{\theta}^{(1)}) + \min_{\boldsymbol{\theta}^{(2)} \in \Theta_B} \sum_{t=\eta+1}^{T} \ell_t(\boldsymbol{\theta}^{(2)}) \right\}, \tag{B.16}$$

*where $\ell_t(\boldsymbol{\theta})$ is the negative log-likelihood given in Equation (2.6). Then it holds with probability at least $1 - (Tn)^{-2}$ that*

$$|\hat{\eta} - \eta| \leq \frac{C p_{lb}^{-2} |E| n d_{\max} \log(Tn)}{\lambda_2^2(L_{\mathcal{G}}) \kappa^2}. \tag{B.17}$$

*Proof.* The estimator $\hat{\eta}$ is the same as the output of the local refinement algorithm. Under the assumption (B.15), the same arguments in the proof of Proposition B.3 can be applied here to show the conclusion.

It should be noted that the estimator $\hat{\eta}$ gives consistent localization because as $T \to \infty$, we have $\mathcal{B}_T \to \infty$ and with high probability,

$$\frac{|\hat{\eta} - \eta|}{\Delta} \leq \frac{C}{\mathcal{B}_T} = o(1).$$

$\square$

**Proposition B.6** (No change point). *Suppose we observe $\{(\mathbf{x}(t), y_t)\}_{t \in [T]}$ following model (2.1) and (2.5) and there is no single change point in $[1, T]$. In addition, assume that*

$$T \geq \mathcal{B}_T \frac{|E| n d_{\max}}{p_{lb}^2 \kappa^2 \lambda_2^2(L_{\mathcal{G}})} \log(Tn), \tag{B.18}$$

*for a diverging sequence $\{\mathcal{B}_T\}_{T \in \mathbb{Z}_+}$. Then it holds with probability at least $1 - (Tn)^{-2}$ that the DP procedure in Algorithm 1 with tuning parameter $\gamma = C_\gamma p_{lb}^{-2} \frac{n d_{\max}}{\lambda_2(L_{\mathcal{G}})} \log(Tn)$ will return an empty set.*

*Proof.* Assume that the output $\hat{\mathcal{P}} = \{\hat{\eta}_k\}_{k \in [\hat{K}]}$ with $\hat{K} \geq 1$. Let $\mathcal{I}_0 = [1, \hat{\eta}_1)$ and $\mathcal{I}_{\hat{K}} = [\hat{\eta}_{\hat{K}}, T]$. When $\hat{K} > 1$, let $\mathcal{I}_k = [\hat{\eta}_{k-1}, \hat{\eta}_k)$ for $k \in [\hat{K} - 1]$. Then by Lemma B.9, with probability at least $1 - (Tn)^{-4}$, we have

$$\sum_{k=0}^{\hat{K}} L(\hat{\boldsymbol{\theta}}(\mathcal{I}_k), \mathcal{I}_k) + \hat{K} C p_{lb}^{-2} \frac{n d_{\max}}{\lambda_2(L_{\mathcal{G}})} \log(Tn) \geq \sum_{k=0}^{\hat{K}} L(\boldsymbol{\theta}^*(\mathcal{I}_k), \mathcal{I}_k)$$

$$= L(\boldsymbol{\theta}^*([1, T]), [1, T]) \geq L(\hat{\boldsymbol{\theta}}([1, T]), [1, T]),$$

which is contradictory to the definition of $\hat{\mathcal{P}}$ as long as $C_\gamma > C$. $\square$

## B.3   Technical lemmas

This section has three parts:

1. Lemma B.7 is a summary of three different cases, and is used in the proof of Proposition B.1.

2. Appendix B.3.1 contains results on the excess risk of $L(\boldsymbol{\theta}(\mathcal{I}), \mathcal{I})$ in four cases.

3. Appendix B.3.2 contains lemmas on some basic concentration properties related to our problem.

**Lemma B.7.** *Given any interval $\mathcal{I} = (s, e] \subset [1, T]$ with integers $s, e$ that contains at most two change points. Under all assumptions above, we have*

1. *If $\mathcal{I}$ contains no change points, then with probability at leat $1 - (Tn)^{-2}$ it holds that*

$$L(\boldsymbol{\theta}^*(\mathcal{I}), \mathcal{I}) \le L(\hat{\boldsymbol{\theta}}(\mathcal{I}), \mathcal{I}) + C p_{lb}^{-2} \frac{n d_{\max}}{\lambda_2(L_{\mathcal{G}})} \log(Tn).$$

2. *If $\mathcal{I}$ contains exactly one change point $\eta_r$ with partition $\mathcal{I}_1 = (s, \eta_r]$ and $\mathcal{I}_2 = (\eta_r, e]$, then with probability at leat $1 - (Tn)^{-2}$ it holds that*

$$L(\boldsymbol{\theta}^*(\mathcal{I}_1), \mathcal{I}_1) + L(\boldsymbol{\theta}^*(\mathcal{I}_2), \mathcal{I}_2) \le L(\hat{\boldsymbol{\theta}}(\mathcal{I}), \mathcal{I}) + C p_{lb}^{-2} \frac{n d_{\max}}{\lambda_2(L_{\mathcal{G}})} \log(Tn).$$

3. *If $\mathcal{I}$ contains exactly two change points $\eta_{r+1}$ and $\eta_{r+2}$ with partition $\mathcal{I}_1 = (s, \eta_{r+1}]$, $\mathcal{I}_2 = (\eta_{r+1}, \eta_{r+2}]$, and $\mathcal{I}_3 = (\eta_{r+2}, e]$, then with probability at leat $1 - (Tn)^{-2}$ it holds that*

$$\sum_{j=1}^{3} L(\boldsymbol{\theta}^*(\mathcal{I}_j), \mathcal{I}_j) \le L(\hat{\boldsymbol{\theta}}(\mathcal{I}), \mathcal{I}) + C p_{lb}^{-2} \frac{n d_{\max}}{\lambda_2(L_{\mathcal{G}})} \log(Tn).$$

*Proof.* Case 1 is guaranteed by Lemma B.9.

For case 3, since $|\mathcal{I}_2| \ge \Delta$, by Assumption 3.1, Lemma B.9 and the definition of $\hat{\boldsymbol{\theta}}$, it holds with probability at least $1 - (Tp)^{-4}$ that

$$L(\boldsymbol{\theta}^*(\mathcal{I}_2), \mathcal{I}_2) \le L(\hat{\boldsymbol{\theta}}(\mathcal{I}_2), \mathcal{I}_2) + C p_{lb}^{-2} \frac{n d_{\max}}{\lambda_2(L_{\mathcal{G}})} \log(Tn) \le L(\hat{\boldsymbol{\theta}}(\mathcal{I}), \mathcal{I}_2) + C p_{lb}^{-2} \frac{n d_{\max}}{\lambda_2(L_{\mathcal{G}})} \log(Tn),$$
(B.19)

where the second inequality is implied by the definition of $\hat{\boldsymbol{\theta}}(\mathcal{I}_2)$.

For $\mathcal{I}_1$, we need to consider two cases. If $|\mathcal{I}_1| \ge \frac{C_0 |E|}{\lambda_2(L_{\mathcal{G}})} \log(Tn)$ where $C_0$ is some fixed absolute constant in the sample size condition in Assumption B.14, then by Lemma B.9, with probability at least $1 - (Tn)^{-4}$ we have

$$L(\boldsymbol{\theta}^*(\mathcal{I}_1), \mathcal{I}_1) \le L(\hat{\boldsymbol{\theta}}(\mathcal{I}_1), \mathcal{I}_1) + C p_{lb}^{-2} \frac{n d_{\max}}{\lambda_2(L_{\mathcal{G}})} \log(Tn) \le L(\hat{\boldsymbol{\theta}}(\mathcal{I}), \mathcal{I}_1) + C p_{lb}^{-2} \frac{n d_{\max}}{\lambda_2(L_{\mathcal{G}})} \log(Tn).$$

Otherwise when $|\mathcal{I}_1| < \frac{C_0 |E|}{\lambda_2(L_{\mathcal{G}})} \log(Tn)$, let $\epsilon_t := y_t - \frac{\exp(\mathbf{x}(t)^\top \boldsymbol{\theta}^*(t))}{1 + \exp(\mathbf{x}(t)^\top \boldsymbol{\theta}^*(t))}$ and we can get

$$L(\boldsymbol{\theta}^*(\mathcal{I}_1), \mathcal{I}_1) - \sum_{t \in \mathcal{I}_1} \ell_t(\hat{\boldsymbol{\theta}}(\mathcal{I}))$$

$$= \sum_{t \in \mathcal{I}_1} \ell_t(\boldsymbol{\theta}^*(\mathcal{I}_1)) - \sum_{t \in \mathcal{I}_1} \ell_t(\hat{\boldsymbol{\theta}}(\mathcal{I}))$$

$$\le \sum_{t \in \mathcal{I}_1} \epsilon_t \mathbf{x}(t)^\top (\hat{\boldsymbol{\theta}}(\mathcal{I}) - \boldsymbol{\theta}^*(\mathcal{I}_1)) - c e^{-2B} [\mathbf{x}(t)^\top (\hat{\boldsymbol{\theta}}(\mathcal{I}) - \boldsymbol{\theta}^*(\mathcal{I}_1))]^2$$

$$\le \frac{e^{2B}}{4c} \sum_{t \in \mathcal{I}_1} [\epsilon_t]^2 \le \frac{e^{2B}}{4c} |\mathcal{I}_1| \le C_1 p_{lb}^{-2} \frac{n d_{\max}}{\lambda_2(L_{\mathcal{G}})} \log(Tn),$$

where the last inequality holds because $|\mathcal{I}_1| < \frac{C_0 |E|}{\lambda_2(L_{\mathcal{G}})} \log(Tn)$ and $|E| \le n d_{\max}$. Similarly, we can show that

$$L(\boldsymbol{\theta}^*(\mathcal{I}_3), \mathcal{I}_3) - \sum_{t \in \mathcal{I}_3} \ell_t(\hat{\boldsymbol{\theta}}(\mathcal{I})) \le C_1 p_{lb}^{-2} \frac{n d_{\max}}{\lambda_2(L_{\mathcal{G}})} \log(Tn).$$

Combining the three facts proves the conclusion for case 3. Similar arguments can be used to prove the conclusion for case 2.                                                                 $\square$

### B.3.1  Excess risk

**Lemma B.8.** *Suppose $\boldsymbol{\theta}, \boldsymbol{\theta}(t)^* \in \Theta_B$, then*

$$\ell_t(\boldsymbol{\theta}) - \ell_t(\boldsymbol{\theta}^*(t)) \ge [\frac{\exp(\mathbf{x}(t)^\top \boldsymbol{\theta}^*(t))}{1 + \exp(\mathbf{x}(t)^\top \boldsymbol{\theta}^*(t))} - y_t] \mathbf{x}(t)^\top (\boldsymbol{\theta} - \boldsymbol{\theta}^*(t)) + c e^{-2B} [\mathbf{x}(t)^\top (\boldsymbol{\theta} - \boldsymbol{\theta}^*(t))]^2.$$
(B.20)

*Proof.* By Taylor expansion,

$$\log(1 + e^{\mathbf{x}(t)^\top \boldsymbol{\theta}}) - \log(1 + e^{\mathbf{x}(t)^\top \boldsymbol{\theta}^*(t)})$$

$$=[\frac{\exp(\mathbf{x}(t)^\top \boldsymbol{\theta}^*(t))}{1 + \exp(\mathbf{x}(t)^\top \boldsymbol{\theta}^*(t))} - y_t]\mathbf{x}(t)^\top(\boldsymbol{\theta} - \boldsymbol{\theta}^*(t)) + \frac{\exp(\mathbf{x}(t)^\top \boldsymbol{\xi})}{[1 + \exp(\mathbf{x}(t)^\top \boldsymbol{\xi})]^2}[\mathbf{x}(t)^\top(\boldsymbol{\theta} - \boldsymbol{\theta}^*(t))]^2$$

$$\geq[\frac{\exp(\mathbf{x}(t)^\top \boldsymbol{\theta}^*(t))}{1 + \exp(\mathbf{x}(t)^\top \boldsymbol{\theta}^*(t))} - y_t]\mathbf{x}(t)^\top(\boldsymbol{\theta} - \boldsymbol{\theta}^*(t)) + \frac{1}{4e^{2B}}[\mathbf{x}(t)^\top(\boldsymbol{\theta} - \boldsymbol{\theta}^*(t))]^2.$$

where $\boldsymbol{\xi}$ is a convex combination of $\boldsymbol{\theta}$ and $\boldsymbol{\theta}^*(t)$. Thus, $\boldsymbol{\xi} \in \Theta_B$ and we also use the facts that $|\mathbf{x}(t)^\top \mathbf{v}| \leq 2B$ for any $\mathbf{v} \in \Theta_B$ and $\frac{e^x}{(1+e^x)^2} \geq \frac{1}{4e^{|x|}}$. $\qquad\square$

**Lemma B.9.** *Assume there is no change points in interval $\mathcal{I}$, then it holds with probability at least $1 - (Tn)^{-4}$ that*

$$L(\hat{\boldsymbol{\theta}}(\mathcal{I}), \mathcal{I}) - L(\boldsymbol{\theta}^*(\mathcal{I}), \mathcal{I}) = \sum_{t \in \mathcal{I}}[\ell_t(\hat{\boldsymbol{\theta}}) - \ell_t(\boldsymbol{\theta}^*)] \geq -Cp_{lb}^{-2}\frac{nd_{\max}}{\lambda_2(L_{\mathcal{G}})}\log(Tn),$$

*where $C$ is a universal constant that is independent of the choice of $\mathcal{I}$.*

*Proof.* Let $\epsilon_t := y_t - \frac{\exp(\mathbf{x}(t)^\top \boldsymbol{\theta}^*(t))}{1 + \exp(\mathbf{x}(t)^\top \boldsymbol{\theta}^*(t))}$. By Lemma B.8, we have

$$L(\boldsymbol{\theta}^*(\mathcal{I}), \mathcal{I}) - L(\hat{\boldsymbol{\theta}}, \mathcal{I})$$

$$\leq \sum_{t \in \mathcal{I}} \epsilon_t \mathbf{x}(t)^\top(\hat{\boldsymbol{\theta}}(\mathcal{I}) - \boldsymbol{\theta}^*(\mathcal{I})) - ce^{-2B}\sum_{t \in \mathcal{I}}[\mathbf{x}(t)^\top(\hat{\boldsymbol{\theta}}(\mathcal{I}) - \boldsymbol{\theta}^*(\mathcal{I}))]^2$$

$$\leq \sum_{t \in \mathcal{I}} \epsilon_t \mathbf{x}(t)^\top(\hat{\boldsymbol{\theta}}(\mathcal{I}) - \boldsymbol{\theta}^*(\mathcal{I})) \qquad\qquad\qquad (B.21)$$

$$\leq \|\hat{\boldsymbol{\theta}}(\mathcal{I}) - \boldsymbol{\theta}^*(\mathcal{I})\|_1 \max_{i \in [p]}|\sum_{t \in \mathcal{I}} \epsilon_t x_i(t)|.$$

When $|\mathcal{I}| \gtrsim \frac{C_0|E|}{\lambda_2(L_{\mathcal{G}})}\log(Tn)$, by Lemma B.16, we have $\|\hat{\boldsymbol{\theta}}(\mathcal{I}) - \boldsymbol{\theta}^*(\mathcal{I})\|_1 \lesssim p_{lb}^{-2}n\sqrt{\frac{|E|\log(Tn)}{|\mathcal{I}|\lambda_2(L_{\mathcal{G}})}}$. Thus, Lemma B.21 ensures that the first term is upper bounded by $C_1 p_{lb}^{-2}n\sqrt{\frac{d_{\max}}{\lambda_2(L_{\mathcal{G}})}}\log(Tn)$ where $C_1$ does not depend on $C_0$. Since $\lambda_2(L_{\mathcal{G}}) \leq 2d_{\max}$, we have

$$C_1 p_{lb}^{-2}n\sqrt{\frac{d_{\max}}{\lambda_2(L_{\mathcal{G}})}}\log(Tn) \leq Cp_{lb}^{-2}\frac{nd_{\max}}{\lambda_2(L_{\mathcal{G}})}\log(Tn).$$

When $|\mathcal{I}| < \frac{C_0|E|}{\lambda_2(L_{\mathcal{G}})}\log(Tn)$, we can bound the difference by

$$L(\boldsymbol{\theta}^*(\mathcal{I}), \mathcal{I}) - L(\hat{\boldsymbol{\theta}}(\mathcal{I}), \mathcal{I})$$

$$\leq \sum_{t \in \mathcal{I}} \epsilon_t \mathbf{x}(t)^\top(\hat{\boldsymbol{\theta}}(\mathcal{I}) - \boldsymbol{\theta}^*(\mathcal{I})) - ce^{-2B}\sum_{t \in \mathcal{I}}[\mathbf{x}(t)^\top(\hat{\boldsymbol{\theta}}(\mathcal{I}) - \boldsymbol{\theta}^*(\mathcal{I}))]^2 \qquad (B.22)$$

$$\leq \frac{e^{2B}}{4c}\sum_{t \in \mathcal{I}} \epsilon_t^2 \leq C_2 p_{lb}^{-2}\frac{C_0|E|}{\lambda_2(L_{\mathcal{G}})}\log(Tn),$$

where we use the fact that $|\epsilon_t| \leq 1$ and $\boldsymbol{\theta}^* \in \Theta_B$ by our assumption, and the basic inequality $ab \leq a^2 + b^2/4$. Since $|E| \leq nd_{\max}$, it holds that

$$C_2 p_{lb}^{-2}\frac{C_0|E|}{\lambda_2(L_{\mathcal{G}})}\log(Tn) \leq Cp_{lb}^{-2}\frac{nd_{\max}}{\lambda_2(L_{\mathcal{G}})}\log(Tn).$$

$\qquad\square$

**Lemma B.10.** *Under all assumptions in Theorem 3.2, let $\mathcal{I} = (s, e] \subset [1, T]$ be any interval containing no change point. Let $\mathcal{I}_1, \mathcal{I}_2$ be two intervals such that $\mathcal{I}_1 \cup \mathcal{I}_2 = \mathcal{I}$. It holds with probability at least $1 - (Tn)^{-4}$ that*

$$L(\hat{\boldsymbol{\theta}}(\mathcal{I}_1), \mathcal{I}_1) + L(\hat{\boldsymbol{\theta}}(\mathcal{I}_2), \mathcal{I}_2) + \gamma \geq L(\boldsymbol{\theta}^*(\mathcal{I}), \mathcal{I}).$$

*Proof.* If $\mathcal{I} < 2C_0 \frac{|E|\log(Tn)}{\lambda_2(L_\mathcal{G})}$, following the same arguments in Lemma B.9 we have that for $i = 1, 2$, with probability at least $1 - (Tn)^{-4}$,

$$L(\boldsymbol{\theta}^*(\mathcal{I}_i), \mathcal{I}_i) - L(\hat{\boldsymbol{\theta}}(\mathcal{I}_i), \mathcal{I}_i) \leq Cp_{lb}^{-2} \frac{nd_{\max}}{\lambda_2(L_\mathcal{G})} \log(Tn).$$

Thus, by the fact that $L(\boldsymbol{\theta}^*(\mathcal{I}), \mathcal{I}) = L(\boldsymbol{\theta}^*(\mathcal{I}_1), \mathcal{I}_1) + L(\boldsymbol{\theta}^*(\mathcal{I}_2), \mathcal{I}_2)$ and $\gamma = C_\gamma(K + 1)p_{lb}^{-2} \frac{nd_{\max}}{\lambda_2(L_\mathcal{G})} \log(Tn)$ with $C_\gamma$ large enough, the conclusion holds.

Now assume $\mathcal{I} > 2C_0 \frac{|E|\log(Tn)}{\lambda_2(L_\mathcal{G})}$. We will prove the lemma by contradiction. Assume that

$$L(\hat{\boldsymbol{\theta}}(\mathcal{I}_1), \mathcal{I}_1) + L(\hat{\boldsymbol{\theta}}(\mathcal{I}_2), \mathcal{I}_2) + \gamma < L(\boldsymbol{\theta}^*(\mathcal{I}), \mathcal{I}).$$

By Lemma B.8, the equation above implies that

$$ce^{-2B} \sum_{t \in \mathcal{I}} [\mathbf{x}(t)^\top \Delta(t)]^2 < -\gamma + \sum_{t \in \mathcal{I}} \epsilon_t \mathbf{x}(t)^\top \Delta(t), \tag{B.23}$$

where $\epsilon_t := y_t - \frac{\exp(\mathbf{x}(t)^\top \boldsymbol{\theta}^*(t))}{1 + \exp(\mathbf{x}(t)^\top \boldsymbol{\theta}^*(t))}$ and $\Delta_i(t) = \hat{\theta}_i(\mathcal{I}) - \theta_i^*(t)$. For (B.23), following the same arguments in the proof of Lemma B.11, we can get that with probability at least $1 - (Tn)^{-4}$,

$$\sum_{t \in \mathcal{I}} \epsilon_t \mathbf{x}(t)^\top \Delta(t) \leq C \sqrt{\frac{nd_{\max}}{|E|} \log(Tn)} \left[ \sum_{t \in \mathcal{I}} \|\Delta(t)\|_2^2 \right]^{\frac{1}{2}}.$$

By Lemma B.17, with probability at least $1 - (Tn)^{-5}$,

$$\sum_{t \in \mathcal{I}} [\mathbf{x}(t)^\top \Delta(t)]^2 \geq \frac{c_1 \lambda_2(L_\mathcal{G})}{|E|} \sum_{t \in \mathcal{I}} \|\Delta(t)\|_2^2.$$

Thus, let $z = \sum_{t \in \mathcal{I}} \|\Delta(t)\|_2^2$ and we have

$$\frac{cc_1 \lambda_2}{e^{2B}|E|} z + \gamma \leq C \sqrt{\frac{nd_{\max}}{|E|} \log(Tn)} \sqrt{z} \leq \frac{C^2 e^{2B} nd_{\max}}{cc_1 \lambda_2} \log(Tn) + \frac{cc_1 \lambda_2}{4e^{2B}|E|} z,$$

which implies that

$$\sum_{t \in \mathcal{I}} \|\Delta(t)\|_2^2 + C_1 \frac{p_{lb}^{-1}|E|}{\lambda_2} \gamma \leq C_2 p_{lb}^{-2} \frac{|E|nd_{\max}}{\lambda_2^2} \log(Tn),$$

which is contradictory to the fact that $\gamma = C_\gamma p_{lb}^{-2}(K + 1) \frac{nd_{\max}}{\lambda_2(L_\mathcal{G})} \log(Tn)$ for sufficiently large constant $C_\gamma$. $\square$

**Lemma B.11.** *For $\mathcal{I} = (s, e) \subset (0, T + 1)$, assume that $\mathcal{I}$ contains only one change point $\eta$. Denote $\mathcal{I}_1 = (s, \eta]$ and $\mathcal{I}_2 = (\eta, e]$. Assume that $\|\boldsymbol{\theta}^*(\mathcal{I}_1) - \boldsymbol{\theta}^*(\mathcal{I}_2)\|_2 = \kappa > 0$. If*

$$L(\hat{\boldsymbol{\theta}}(\mathcal{I}), \mathcal{I}) \leq L(\boldsymbol{\theta}^*(\mathcal{I}_1), \mathcal{I}_1) + L(\boldsymbol{\theta}^*(\mathcal{I}_2), \mathcal{I}_2) + \gamma,$$

*then with probability at least $1 - (Tn)^{-4}$, there exists an absolute constant $C > 0$ such that*

$$\min\{|\mathcal{I}_1|, |\mathcal{I}_2|\} \leq C \frac{p_{lb}^{-2}|E|}{\kappa^2 \lambda_2(L_\mathcal{G})} [\gamma + \frac{nd_{\max}}{\lambda_2(L_\mathcal{G})} \log(Tn)].$$

*Proof.* Without loss of generality, assume $|\mathcal{I}_1| \geq |\mathcal{I}_2|$. If $|\mathcal{I}_2| < C_0 \frac{|E|\log(Tn)}{\lambda_2(L_\mathcal{G})}$ then the conclusion holds automatically, where $C_0$ is the constant in Lemma B.16 and Lemma B.19, since we can set $C_\gamma$ to be sufficiently large (notice that in the worst case, $\kappa^2$ can be as large as $nB^2$). Thus, in what follows we can assume $|\mathcal{I}_2| \geq C_0 \frac{|E|\log(Tn)}{\lambda_2(L_\mathcal{G})}$. Let $\epsilon_t := y_t - \frac{\exp(\mathbf{x}(t)^\top \boldsymbol{\theta}^*(t))}{1 + \exp(\mathbf{x}(t)^\top \boldsymbol{\theta}^*(t))}$ and $\Delta_i(t) = \hat{\theta}_i(\mathcal{I}) - \theta_i^*(t)$. By the condition of the lemma and Lemma B.8, we have

$$ce^{-2B} \sum_{t \in \mathcal{I}} [\mathbf{x}(t)^\top \Delta(t)]^2 \leq \gamma + \sum_{t \in \mathcal{I}} \sum_{i \in [n]} \epsilon_t x_i(t) \Delta_i(t).$$

Lemma B.19 implies that with probability at least $1 - (Tn)^{-4}$, the term on the right hand side satisfies

$$
\sum_{t \in \mathcal{I}} \sum_{i \in [n]} \epsilon_t x_i(t)(\hat{\theta}_i - \theta_i^*(t))
$$

$$
\leq \sup_{i \in [n]} \left| \frac{\sum_{t \in \mathcal{I}} \epsilon_t x_i(t)(\hat{\theta}_i - \theta_i^*(t))}{\sqrt{\sum_{t \in \mathcal{I}} (\hat{\theta}_i - \theta_i^*(t))^2}} \right| \sum_{i \in [n]} \sqrt{\sum_{t \in \mathcal{I}} (\hat{\theta}_i - \theta_i^*(t))^2}
$$

$$
\leq C \sqrt{\frac{d_{\max}}{|E|} \log(Tn)} \sum_{i \in [n]} \sqrt{\sum_{t \in \mathcal{I}} (\hat{\theta}_i - \theta_i^*(t))^2} \leq C \sqrt{\frac{n d_{\max}}{|E|} \log(Tn)} \sqrt{\sum_{t \in \mathcal{I}} \|\Delta(t)\|_2^2}.
$$

By Lemma B.17, $\sum_{t \in \mathcal{I}_i} [\mathbf{x}(t)^\top \Delta(t)]^2 \geq \frac{c_1 \lambda_2(L_\mathcal{G})}{|E|} \sum_{t \in \mathcal{I}_i} \|\Delta(t)\|_2^2$ with probability at least $1 - (Tn)^{-5}$ for $i = 1, 2$. Therefore, letting $z = \sum_{t \in \mathcal{I}} \|\Delta(t)\|_2^2$, we have

$$
cc_1 \frac{\lambda_2(L_\mathcal{G})}{e^{2B}|E|} z \leq \gamma + c_2 \sqrt{\frac{d_{\max}}{|E|} \log(Tn)} \sqrt{z}.
$$

Solving the inequality above gives

$$
\sum_{t \in \mathcal{I}} \|\Delta(t)\|_2^2 \leq C p_{lb}^{-2} \frac{|E|}{\lambda_2(L_\mathcal{G})} [\gamma + \frac{n d_{\max}}{\lambda_2(L_\mathcal{G})} \log(Tn)],
$$

where $C$ is a universal constant that only depends on $c, c_1, c_2$. Since $\sum_{t \in \mathcal{I}} \|\Delta(t)\|_2^2 \geq \frac{|\mathcal{I}_1||\mathcal{I}_2|}{|\mathcal{I}|} \kappa^2 \geq \frac{\kappa^2}{2} |\mathcal{I}_2|$, we have $|\mathcal{I}_2| \leq \frac{2C}{\kappa^2} p_{lb}^{-2} \frac{|E|}{\lambda_2(L_\mathcal{G})} [\gamma + \frac{n d_{\max}}{\lambda_2(L_\mathcal{G})} \log(Tn)]$. $\square$

**Lemma B.12.** *Under all assumptions in Theorem 3.2, let $\mathcal{I} = (s, e] \subset [1, T]$ be any interval containing exactly two change points $\eta_{r+1}$ and $\eta_{r+2}$, $\mathcal{I}_1 = (e, \eta_{r+1}]$, $\mathcal{I}_2 = (\eta_{r+1}, \eta_{r+2}]$, and $I_3 = (\eta_{r+2}, e]$. Let $\kappa_i = \|\boldsymbol{\theta}^*(\mathcal{I}_i) - \boldsymbol{\theta}^*(\mathcal{I}_{i+1})\|_2$ for $i = 1, 2$ and $\kappa = \min\{\kappa_1, \kappa_2\}$. If*

$$
L(\hat{\boldsymbol{\theta}}(\mathcal{I}), \mathcal{I}) \leq \sum_{i=1}^{3} L(\boldsymbol{\theta}^*(\mathcal{I}_i), \mathcal{I}_i) + 2\gamma,
$$

*then it holds with probability at least $1 - (Tn)^{-4}$ that*

$$
\max\{|\mathcal{I}_1|, |\mathcal{I}_3|\} \leq C p_{lb}^{-2} \frac{|E|}{\lambda_2(L_\mathcal{G})} [\gamma + \frac{n d_{\max}}{\lambda_2(L_\mathcal{G})} \log(Tn)].
$$

*Proof.* Without loss of generality, we assume $|\mathcal{I}_1| \geq |\mathcal{I}_3|$. There are three possible cases: 1. $|\mathcal{I}_1| \leq C_0 \frac{|E| \log(Tn)}{\lambda_2(L_\mathcal{G})}$, 2. $|\mathcal{I}_3| \geq C_0 \frac{|E| \log(Tn)}{\lambda_2(L_\mathcal{G})}$, and 3. $|\mathcal{I}_1| \geq C_0 \frac{|E| \log(Tn)}{\lambda_2(L_\mathcal{G})} \geq |\mathcal{I}_3|$ where $C_0$ is the constant in Lemma B.16 and Lemma B.19. In case 1 the conclusion holds immediately since we can set $C_\gamma$ to be large enough. In case 2, the condition in the lemma implies that

$$
ce^{-2B} \sum_{t \in \mathcal{I}} [\mathbf{x}(t)^\top \Delta(t)]^2 \leq 2\gamma + \sum_{t \in \mathcal{I}} \epsilon_t \mathbf{x}(t)^\top \Delta(t),
$$

where $\epsilon_t := y_t - \frac{\exp(\mathbf{x}(t)^\top \boldsymbol{\theta}^*(t))}{1 + \exp(\mathbf{x}(t)^\top \boldsymbol{\theta}^*(t))}$ and $\Delta_i(t) = \hat{\theta}_i(\mathcal{I}) - \theta_i^*(t)$.

For the term involving $\epsilon_t$, following the same arguments in the proof of Lemma B.11, we can get that with probability at least $1 - (Tn)^{-4}$,

$$
\sum_{t \in \mathcal{I}} \epsilon_t \mathbf{x}(t)^\top \Delta(t) \leq C \sqrt{\frac{n d_{\max}}{|E|} \log(Tn)} \left[ \sum_{t \in \mathcal{I}} \|\Delta(t)\|_2^2 \right]^{\frac{1}{2}}.
$$

Let $z = \sum_{t \in \mathcal{I}} \|\Delta(t)\|_2^2$. By Lemma B.17, $\sum_{t \in \mathcal{I}} [\mathbf{x}(t)^\top \Delta(t)]^2 \geq \frac{c_1 \lambda_2(L_\mathcal{G})}{|E|} \sum_{t \in \mathcal{I}} \|\Delta(t)\|_2^2$ with probability at least $1 - (Tn)^{-5}$, and thus we have

$$
\frac{cc_1 \lambda_2(L_\mathcal{G})}{e^{2B}|E|} z \leq C \sqrt{\frac{n d_{\max}}{|E|} \log(Tn)} \sqrt{z} + 2\gamma,
$$

which implies that

$$\sum_{t \in \mathcal{I}} \|\Delta(t)\|_2^2 \le C_1 p_{lb}^{-2} \frac{|E| n d_{\max}}{\lambda_2^2(L_\mathcal{G})} \log(Tn) + C_2 \frac{e^{2B}|E|}{\lambda_2(L_\mathcal{G})} \gamma.$$

Denote $\tilde{\mathcal{I}}$ as the shorter one of $|\mathcal{I}_1|$ and $|\mathcal{I}_2|$. The left hand can be lowered bounded by

$$\sum_{t \in \mathcal{I}} \|\Delta(t)\|_2^2 \ge \sum_{t \in \mathcal{I}_1 \cup \mathcal{I}_2} \|\Delta(t)\|_2^2 \ge \frac{|\mathcal{I}_1||\mathcal{I}_2|}{|\mathcal{I}_1| + |\mathcal{I}_2|} \kappa^2 \ge \frac{|\tilde{\mathcal{I}}|}{2} \kappa^2.$$

If $|\mathcal{I}_2| < |\mathcal{I}_1|$, then we have

$$\frac{|\mathcal{I}_2|}{2} \kappa^2 \le C_1 p_{lb}^{-2} \frac{|E| n d_{\max}}{\lambda_2^2(L_\mathcal{G})} \log(Tn) + C_2 p_{lb}^{-1} \frac{|E|}{\lambda_2(L_\mathcal{G})} \gamma,$$

which leads to the bound

$$|\mathcal{I}_2| \lesssim \frac{p_{lb}^{-2}|E|}{\kappa^2 \lambda_2(L_\mathcal{G})} [\gamma + \frac{n d_{\max}}{\lambda_2(L_\mathcal{G})} \log(Tn)],$$

and is contradictory to the assumption that $\Delta \ge \mathcal{B}_T p_{lb}^{-4} K \frac{|E| n d_{\max}}{\kappa^2 \lambda_2(L_\mathcal{G})} \log(Tn)$ in Assumption 3.1 because of the definition $\gamma = C_\gamma p_{lb}^{-2}(K+1) \frac{n d_{\max}}{\lambda_2(L_\mathcal{G})} \log(Tn)$. Therefore, we have $|\mathcal{I}_2| \ge |\mathcal{I}_1|$ and by the same arguments,

$$|\mathcal{I}_1| \lesssim \frac{p_{lb}^{-2}|E|}{\kappa^2 \lambda_2(L_\mathcal{G})} [\gamma + \frac{n d_{\max}}{\lambda_2(L_\mathcal{G})} \log(Tn)].$$

Since we assume $|\mathcal{I}_3| \le |\mathcal{I}_1|$, the desired bound holds.

In case 3, we only need to prove that $|\mathcal{I}_1| \le C \frac{p_{lb}^{-2}|E|}{\kappa^2 \lambda_2(L_\mathcal{G})} [\gamma + \frac{n d_{\max}}{\lambda_2(L_\mathcal{G})} \log(Tn)]$. Following the same arguments for Equation (B.22), we can get that with probability at least $1 - (Tn)^{-5}$,

$$L(\boldsymbol{\theta}^*(\mathcal{I}_3), \mathcal{I}_3) - L(\hat{\boldsymbol{\theta}}(\mathcal{I}), \mathcal{I}_3) \le C p_{lb}^{-2} \frac{n d_{\max}}{\lambda_2(L_\mathcal{G})} \log(Tn) \le \gamma/3.$$

Therefore, by the condition of the lemma, we have

$$L(\hat{\boldsymbol{\theta}}(\mathcal{I}), \mathcal{I}_1 \cup \mathcal{I}_2) \le \sum_{i=1}^{2} L(\boldsymbol{\theta}^*(\mathcal{I}_i), \mathcal{I}_i) + \frac{7}{3} \gamma.$$

Since $\mathcal{I}_1 \cup \mathcal{I}_2$ only contains 1 true change point, the conclusion can be shown by the same arguments of Lemma B.11. $\square$

**Lemma B.13.** *Under all assumptions in Theorem 3.2, let $\mathcal{I} = (s, e] \subset [1, T]$ be any interval containing $J \ge 3$ change points $\eta_{r+1}, \cdots, r+J$. Let $\mathcal{I}_1 = (e, \eta_{r+1}]$, $\mathcal{I}_j = (\eta_{r+j-1}, \eta_{r+j}]$ for $j = 2, \cdots, J$, and $\mathcal{I}_{J+1} = (\eta_{r+J}, e]$. Also let $\kappa_j = \|\boldsymbol{\theta}^*(\mathcal{I}_j) - \boldsymbol{\theta}^*(\mathcal{I}_{j+1})\|_2$ for $j \in [J]$ and $\kappa = \min_{j \in [J]}\{\kappa_j\}$. Then it holds with probability at least $1 - (Tn)^{-4}$ that*

$$L(\hat{\boldsymbol{\theta}}(\mathcal{I}), \mathcal{I}) > \sum_{j=1}^{J+1} L(\boldsymbol{\theta}^*(\mathcal{I}_j), \mathcal{I}_j) + J\gamma,$$

*Proof.* Without loss of generality, assume that $|\mathcal{I}_1| \ge |\mathcal{I}_{J+1}|$. Similar to Lemma B.12, there are three cases: 1. $|\mathcal{I}_1| \le C_0 \frac{|E| \log(Tn)}{\lambda_2(L_\mathcal{G})}$, 2. $|\mathcal{I}_{J+1}| \ge C_0 \frac{|E| \log(Tn)}{\lambda_2(L_\mathcal{G})}$, and 3. $|\mathcal{I}_1| \ge C_0 \frac{|E| \log(Tn)}{\lambda_2(L_\mathcal{G})} \ge |\mathcal{I}_{J+1}|$ where $C_0$ is the constant in Lemma B.16 and Lemma B.19. In case 2, we prove the conclusion by contradiction. Assume that

$$L(\hat{\boldsymbol{\theta}}(\mathcal{I}), \mathcal{I}) \le \sum_{j=1}^{J+1} L(\boldsymbol{\theta}^*(\mathcal{I}_j), \mathcal{I}_j) + J\gamma$$

We have

$$ce^{-2B} \sum_{t \in \mathcal{I}} [\mathbf{x}(t)^\top \Delta(t)]^2 \le J\gamma + \sum_{t \in \mathcal{I}} \epsilon_t \mathbf{x}(t)^\top \Delta(t),$$

where $\epsilon_t := y_t - \frac{\exp(\mathbf{x}(t)^\top \boldsymbol{\theta}^*(t))}{1+\exp(\mathbf{x}(t)^\top \boldsymbol{\theta}^*(t))}$ and $\Delta_i(t) = \hat{\theta}_i(I) - \theta_i^*(t)$.

For the term that contains $\epsilon_t$, we can bound it as

$$\sum_{t \in \mathcal{I}} \epsilon_t \mathbf{x}(t)^\top \Delta(t) \leq C \sqrt{\frac{n d_{\max}}{|E|} \log(Tn)} \left[ \sum_{t \in \mathcal{I}} \|\Delta(t)\|_2^2 \right]^{\frac{1}{2}},$$

with probability at least $1 - (Tn)^{-4}$. Combining the bounds on both terms and use Lemma B.17 lead to a similar inequality in Lemma B.12 whose solution gives us

$$\sum_{t \in \mathcal{I}} \|\Delta(t)\|_2^2 \leq C_1 p_{lb}^{-2} \frac{|E| n d_{\max}}{\lambda_2^2(L_\mathcal{G})} \log(Tn) + C_2 J \frac{e^{2B}|E|}{\lambda_2(L_\mathcal{G})} \gamma.$$

By definition we know that for $1 \leq j \leq J$, $|\mathcal{I}_j| \geq \Delta$ and thus,

$$\sum_{t \in \mathcal{I}} \|\Delta(t)\|_2^2 \geq \sum_{j=1}^{J} \sum_{t \in \mathcal{I}_j} \|\Delta(t)\|_2^2$$

$$\geq \sum_{j=1}^{J-1} \frac{1}{2} [\sum_{t \in \mathcal{I}_j} \|\Delta(t)\|_2^2 + \sum_{t \in \mathcal{I}_{j+1}} \|\Delta(t)\|_2^2]$$

$$\geq \sum_{j=1}^{J-1} \frac{1}{2} \cdot \frac{|\mathcal{I}_j||\mathcal{I}_{j+1}|}{|\mathcal{I}_j| + |\mathcal{I}_{j+1}|} \kappa^2$$

$$\geq \frac{1}{4}(J-1)\Delta\kappa^2.$$

Therefore, we have

$$\Delta\kappa^2 \leq C_3 p_{lb}^{-2} \frac{|E|}{\lambda_2(L_\mathcal{G})} [\gamma + \frac{n d_{\max}}{J\lambda_2(L_\mathcal{G})} \log(Tn)].$$

Since we assume $|\mathcal{I}_2| \geq |\mathcal{I}_3|$, the inequality above contradicts to the assumption that $\Delta\kappa^2 \geq \mathcal{B}_T p_{lb}^{-4} K \frac{|E| n d_{\max}}{\lambda_2^2(L_\mathcal{G})} \log(Tn)$ in Assumption 3.1.

In case 1, following the same arguments of Equation (B.22), we can get that for $j = 1, J+1$, with probability at least $1 - (Tn)^{-5}$,

$$L(\boldsymbol{\theta}^*(\mathcal{I}_j), \mathcal{I}_j) - L(\hat{\boldsymbol{\theta}}(\mathcal{I}), \mathcal{I}_j) \leq C p_{lb}^{-2} \frac{C_0 |E|}{\lambda_2(L_\mathcal{G})} \log(Tn) \leq \gamma/3.$$

Similar to case 2, we assume that

$$L(\hat{\boldsymbol{\theta}}(\mathcal{I}), \mathcal{I}) \leq \sum_{j=1}^{J+1} L(\boldsymbol{\theta}^*(\mathcal{I}_j), \mathcal{I}_j) + J\gamma.$$

Therefore,

$$\sum_{j=2}^{J} L(\hat{\boldsymbol{\theta}}(\mathcal{I}), \mathcal{I}_j) \leq \sum_{j=2}^{J} L(\boldsymbol{\theta}^*(\mathcal{I}_j), \mathcal{I}_j) + (J + \frac{2}{3})\gamma.$$

When $J = 3$, following same arguments in Lemma B.12, we lead to a contradiction that $\Delta \leq C p_{lb}^{-2} \frac{|E|}{\kappa^2 \lambda_2} [\gamma + \frac{n d_{\max}}{\lambda_2} \log(Tn)]$. When $J > 3$, we can get the same contradiction by the same arguments for case 2 in this lemma. Case 3 can be handled in a similar manner.                $\square$

### B.3.2   Basic concentrations

First we introduce some results on the empirical risk minimizer of the Bradley-Terry model, which is defined by the constraint MLE

$$\hat{\boldsymbol{\theta}} = \arg\min_{\theta \in \Theta_B} \sum_{i \in [m]} \ell_i(\boldsymbol{\theta}). \tag{B.24}$$

**Assumption B.14.** Assume that $(\mathbf{x}(t), y_t)_{t \in [m]}$ are i.i.d. observations generated from model (2.5) with $\boldsymbol{\theta}^*(t) = \boldsymbol{\theta}^* \in \Theta_B$ being a constant vector and (2.1) and the sample size $m$ satisfies $m \geq C_0 \frac{|E| \log n}{\lambda_2(L_{\mathcal{G}})}$.

Denote $G(\mathcal{G}, m)$ as the (weighted) random graph constructed by randomly sampling $m$ edges with replacement from a fixed symmetric, undirected, and binary graph $\mathcal{G}([n], E)$ of $n$ nodes.

**Lemma B.15** (Laplacian, general graph)**.** *Let $A$ be a (weighted) adjacency matrix sampled from the random graph model $G(\mathcal{G}, m)$ and $L_A = D - A$ be the Laplacian matrix. Denote the eigenvalues of a Laplacian matrix $L$ as $0 = \lambda_1(L) \leq \lambda_2(L) \leq \cdots \leq \lambda_n(L)$ for $L = L_A, L_{\mathcal{G}}$. Suppose $m \geq C_0 \frac{|E| \log n}{\lambda_2(L_{\mathcal{G}})}$ for some sufficiently large constant $C_0 > 0$, then with probability at least $1 - O(n^{-10})$ we have*

$$\frac{m\lambda_2(L_{\mathcal{G}})}{2|E|} \leq \lambda_2(L_A) \leq \lambda_n(L_A) \leq \frac{3m\lambda_n(L_{\mathcal{G}})}{|E|}. \tag{B.25}$$

*Proof.* Consider a partial isometry matrix $R \in \mathbb{R}^{(n-1) \times n}$ that satisfies $RR^\top = I_{n-1}$ and $R\mathbf{1}_n = 0$. By basic algebra we know that $\operatorname{rank}(R) = n - 1$ and $\{R^\top v : v \in \mathbb{R}^{n-1}\} = \{a\mathbf{1}_n : a \in \mathbb{R}\}^\perp$. Consider $Y = RL_{\mathcal{G}}R^\top$, then the eigenvalues $\{\lambda_i(L_{\mathcal{G}})\}_{i=2}^n$ are the same as eigenvalues of $Y$. Since $\mathbb{E}[Y] = \frac{m}{|E|} RL_{\mathcal{G}}R^\top$, by matrix Chernoff inequality (e.g., Theorem 5.1.1 in Tropp (2015)), we have

$$\mathbb{P}(\lambda_2(L_A) \leq \frac{m\lambda_2(L_{\mathcal{G}})}{2|E|}) = \mathbb{P}(\lambda_{\min}(Y) \leq \frac{m\lambda_2(L_{\mathcal{G}})}{2|E|}) \leq n \exp(-\frac{m\lambda_2(L_{\mathcal{G}})}{8|E|}) \leq n^{-10} \tag{B.26}$$

for $m \geq C_0 \frac{|E| \log n}{\lambda_2(L_{\mathcal{G}})}$ where $C_0$ is a sufficiently large constant. Similarly, we can show that $\lambda_n(L_A) < \frac{3m\lambda_n(L_{\mathcal{G}})}{|E|}$ with probability at least $1 - n^{-10}$. $\square$

**Lemma B.16** (Estimation of BTL, general graph)**.** *Under Assumption B.14, for the MLE $\hat{\boldsymbol{\theta}}$ defined in Equation (B.24), with probability at least $1 - O(n^{-10})$ we have*

$$\|\hat{\boldsymbol{\theta}} - \boldsymbol{\theta}^*\|_2 \leq Cp_{lb}^{-2} \sqrt{\frac{n|E| \log n}{m\lambda_2(L_{\mathcal{G}})}}, \quad \|\hat{\boldsymbol{\theta}} - \boldsymbol{\theta}^*\|_1 \leq Cp_{lb}^{-2} n \sqrt{\frac{|E| \log n}{m\lambda_2(L_{\mathcal{G}})}}. \tag{B.27}$$

*Proof.* The first inequality is a corollary of Theorem 2 in Shah et al. (2016) and Lemma B.17. Specifically, Shah et al. (2016) ensures that with probability at least $1 - O(n^{-12})$,

$$\|\hat{\boldsymbol{\theta}} - \boldsymbol{\theta}^*\|_2^2 \leq Cp_{lb}^{-4} \frac{n \log(n)}{\lambda_2(L_A)}.$$

By Equation (B.26), $\lambda_2(L_A) \geq \frac{m\lambda_2(L_{\mathcal{G}})}{2|E|}$ with probability at least $1 - O(n^{-12})$, so a union bound leads to the conclusion. The second inequality is implied by $\|x\|_1 \leq \sqrt{n}\|x\|_2$ for any $x \in \mathbb{R}^n$. $\square$

As a special case, the random graph model $G(n, m)$ generates random graphs with the vertex set $[n]$ and $m$ edges randomly sampled from the full edge set $E_{full} = \{(i,j) : 1 \leq i < j \leq n\}$. Lemma B.17 gives high probability bounds for the spectra of random graphs following $G(n, m)$.

**Lemma B.17** (Laplacian, complete graph)**.** *Let $A$ be a (weighted) adjacency matrix sampled from the random graph model $G(n, m)$ and $L_A = D - A$ be the Laplacian matrix. Denote the eigenvalues of $L_A$ as $0 = \lambda_1 \leq \lambda_2 \leq \cdots \leq \lambda_n$. Suppose $m \geq C_0 n \log n$ for some sufficiently large constant $C_0 > 0$, then with probability at least $1 - O(n^{-10})$ we have*

$$\frac{m}{n} \leq \lambda_2(L_A) \leq \lambda_n(L_A) \leq \frac{4m}{n}. \tag{B.28}$$

*Proof.* Consider a partial isometry matrix $R \in \mathbb{R}^{(n-1) \times n}$ that satisfies $RR^\top = I_{n-1}$ and $R\mathbf{1}_n = 0$. By basic algebra we know that $\operatorname{rank}(R) = n - 1$ and $\{R^\top v : v \in \mathbb{R}^{n-1}\} = \{a\mathbf{1}_n : a \in \mathbb{R}\}^\perp$. Consider $Y = RL_A R^\top$, then the eigenvalues $\{\lambda_i\}_{i=2}^n$ are the same as eigenvalues of $Y$. Since $\mathbb{E}[Y] = \frac{2m}{n-1} I_{n-1}$, by matrix Chernoff inequality (e.g., Theorem 5.1.1 in Tropp (2015)), we have

$$\mathbb{P}(\lambda_2(L_A) \leq \frac{m}{n-1}) = \mathbb{P}(\lambda_{\min}(Y) \leq \frac{m}{n-1}) \leq (n-1)e^{-\frac{m}{8(n-1)}} \leq n^{-10} \tag{B.29}$$

for $m \geq C_0 n \log n$ where $C_0$ is a sufficiently large constant. Similarly, we can show that $\lambda_n(L_A) \leq 4m/n$ with probability at least $1 - n^{-10}$. $\qquad\square$

**Lemma B.18** (Estimation of BTL, complete graph). *Under Assumption B.14, for the MLE $\hat{\boldsymbol{\theta}}$ defined in Equation* (B.24)*, with probability at least $1 - O(n^{-10})$ we have*

$$\|\hat{\boldsymbol{\theta}} - \boldsymbol{\theta}^*\|_2 \leq C p_{lb}^{-2} n \sqrt{\frac{\log n}{m}}, \quad \|\hat{\boldsymbol{\theta}} - \boldsymbol{\theta}^*\|_1 \leq C p_{lb}^{-2} n^{3/2} \sqrt{\frac{\log n}{m}}. \tag{B.30}$$

*Proof.* The first inequality is a corollary of Theorem 2 in Shah et al. (2016) and Lemma B.17. Specifically, Shah et al. (2016) ensures that with probability at least $1 - O(n^{-12})$,

$$\|\hat{\boldsymbol{\theta}} - \boldsymbol{\theta}^*\|_2^2 \leq C p_{lb}^{-4} \frac{n \log(n)}{\lambda_2(L_A)}.$$

By Equation (B.26), $\lambda_2(L_A) \geq m/n$ with probability at least $1 - O(n^{-12})$, so a union bound leads to the conclusion. The second inequality is implied by $\|x\|_1 \leq \sqrt{n}\|x\|_2$ for any $x \in \mathbb{R}^n$. $\qquad\square$

In what follows, we prove some concentration properties related to $\epsilon_t := y_t - \frac{\exp(\mathbf{x}(t)^\top \boldsymbol{\theta}^*(t))}{1+\exp(\mathbf{x}(t)^\top \boldsymbol{\theta}^*(t))}$.

**Lemma B.19.** *Under all assumptions in Theorem 3.2, let $\mathcal{I} = [1, T]$ be an integer interval such that $|\mathcal{I}| \geq c_0(R+1)\frac{|E|}{\lambda_2(L_{\mathcal{G}})} \log(Tn)$ and $R$ be a fixed integer. Denote $S_{\mathcal{I},R} = \{\mathbf{v} \in \mathbb{R}^{|\mathcal{I}|} : \|\mathbf{v}\|_2 = 1, \|D\mathbf{v}\|_0 = R, \min\{k : v_j \neq v_{j+k}\} \geq c_0 \frac{|E|}{\lambda_2(L_{\mathcal{G}})} \log(Tn)\}$ and $\epsilon_t := y_t - \frac{\exp(\mathbf{x}(t)^\top \boldsymbol{\theta}^*(t))}{1+\exp(\mathbf{x}(t)^\top \boldsymbol{\theta}^*(t))}$. Then for some sufficiently large constant $C$, it holds with probability at least $1 - (Tn)^{-2R-10}$ that*

$$\max_{i \in [p]} \sup_{\mathbf{v} \in S_{\mathcal{I},R}} \sum_{t \in I} v_t \epsilon_t x_i(t) \leq C \sqrt{\frac{d_{\max} R \log(Tn)}{|E|}}.$$

*Proof.* Since $\|D\mathbf{v}\|_0 = R$, $\{v_t\}$ is piece-wise constant over $\mathcal{I} = [1, T]$ and has $R$ change points that have at most $\binom{T}{R}$ possible choices of locations. Let $\{\eta_k\}_{k \in [R]}$ be the change points of $\{v_t\}$ and $\mathcal{S}(\{\eta_k\}_{k \in [R]})$ the linear subspace of $\mathcal{R}^{|\mathcal{I}|}$ that contains all piecewise-linear sequences over $\mathcal{I}$ whose change points are $\{\eta_k\}_{k \in [R]}$. Let $\mathcal{N}_\delta(\{\eta_k\}_{k \in [R]})$ be a $\delta$-net of $\mathcal{S}(\{\eta_k\}_{k \in [R]}) \cap \mathcal{S}^{|\mathcal{I}|}$ where $\mathcal{S}^{|\mathcal{I}|}$ is the unit sphere in $\mathcal{R}^{|\mathcal{I}|}$. By Lemma 4.1 in Pollard (1990), since $\mathcal{S}(\{\eta_k\}_{k \in [R]})$ is an affine space with dimension $R + 1$, we can pick a $\delta$-net $\mathcal{N}_\delta(\{\eta_k\}_{k \in [R]})$ such that $|\mathcal{N}_\delta(\{\eta_k\}_{k \in [R]})| \leq (\frac{3}{\delta})^{R+1}$.

Taking $\delta = \frac{1}{|\mathcal{I}|}$, then for any fixed $i \in [n]$ and fixed set of change points $\{\eta_k\}_{k \in [R]}$, we have

$$\mathbb{P}\left[\sup_{\mathbf{v} \in \mathcal{S}_{\mathcal{I},R}} \sum_{t \in \mathcal{I}} v_t \epsilon_t x_i(t) \geq C \sqrt{d_{\max} R \log(Tn)/|E|}\right]$$

$$\leq \mathbb{P}\left[\sup_{\mathbf{u} \in \mathcal{N}_{1/|\mathcal{I}|}(\{\eta_k\}_{k \in [R]})} |\sum_{t \in \mathcal{I}} u_t \epsilon_t x_i(t)| + \sup_{\mathbf{v} \in \mathcal{S}_{\mathcal{I},R}} \inf_{\mathbf{u} \in \mathcal{N}_{1/|\mathcal{I}|}} |\sum_{t \in \mathcal{I}} (v_t - u_t) \epsilon_t x_i(t)| \geq C \sqrt{d_{\max} R \log(Tn)/|E|}\right]$$

$$\leq \mathbb{P}\left[\sup_{\mathbf{u} \in \mathcal{N}_{1/|\mathcal{I}|}(\{\eta_k\}_{k \in [R]})} |\sum_{t \in \mathcal{I}} u_t \epsilon_t x_i(t)| + \sup_{\mathbf{v}} \inf_{\mathbf{u}} \|\mathbf{v} - \mathbf{u}\|_1 \max_{t \in \mathcal{I}} |\epsilon_t x_i(t)| \geq C \sqrt{d_{\max} R \log(Tn)/|E|}\right]$$

$$\leq \mathbb{P}\left[\sup_{\mathbf{u} \in \mathcal{N}_{1/|\mathcal{I}|}(\{\eta_k\}_{k \in [R]})} |\sum_{t \in \mathcal{I}} u_t \epsilon_t x_i(t)| + \frac{\sqrt{|\mathcal{I}|}}{|\mathcal{I}|} \cdot \max_{t \in \mathcal{I}} |\epsilon_t x_i(t)| \geq C \sqrt{d_{\max} R \log(Tn)/|E|}\right]$$

$$\leq \mathbb{P}\left[\sup_{\mathbf{u} \in \mathcal{N}_{1/|\mathcal{I}|}(\{\eta_k\}_{k \in [R]})} |\sum_{t \in \mathcal{I}} u_t \epsilon_t x_i(t)| \geq C \sqrt{d_{\max} R \log(Tn)/|E|}\right]$$

$$\times \mathbb{P}\left[\max_{t \in \mathcal{I}} |\epsilon_t x_i(t)| < C \sqrt{d_{\max} R|\mathcal{I}| \log(Tn)/|E|}\right] + \mathbb{P}\left[\max_{t \in \mathcal{I}} |\epsilon_t x_i(t)| \geq C \sqrt{d_{\max} R|\mathcal{I}| \log(Tn)/|E|}\right]$$

Since $\|\mathbf{x}(t)\|_\infty \leq 1$ and $|\epsilon_t| \leq 2$ under Model (2.5), we can make $C$ sufficiently large so that $\mathbb{P}\left[\max_{t \in \mathcal{I}} |\epsilon_t x_i(t)| \geq C\sqrt{R|\mathcal{I}|\log(Tn)/n}\right] = 0$. Therefore,

$$\mathbb{P}\left[\sup_{\mathbf{v} \in \mathcal{S}_{\mathcal{I},R}} \sum_{t \in \mathcal{I}} v_t \epsilon_t x_i(t) \geq C\sqrt{d_{\max}R\log(Tn)/|E|}\right]$$

$$\leq (3|\mathcal{I}|)^{R+1} \sup_{\mathbf{u} \in \mathcal{N}_{1/|\mathcal{I}|}(\{\eta_k\}_{k \in [R]})} \mathbb{P}\left[|\sum_{t \in \mathcal{I}} u_t \epsilon_t x_i(t)| \geq C\sqrt{d_{\max}R\log(Tn)/|E|}\right]$$

$$\leq (3|\mathcal{I}|)^{R+1} \times \max\{2\exp\left[-\frac{CR\log(Tn)}{\sum_{t \in \mathcal{I}} u_t^2}\right], (Tn)^{-3R-12}\}$$

$$\leq C_2 \exp(-C_2 R\log(Tn) + R\log(3|\mathcal{I}|)),$$

where in the second inequality we use Lemma B.20. Therefore, for the given interval $\mathcal{I} \subset [1, T]$, it holds that

$$\mathbb{P}(\mathcal{B}_R(\mathcal{I})) \leq \binom{T}{R} C_2 \exp(-C_3 R\log(Tn)) \leq (Tn)^{-2R-10},$$

where the event $\mathcal{B}_R(\mathcal{I})) := \{\max_{i \in [n]} \sup_{v \in \mathcal{S}_{\mathcal{I},R}} \sum_{t \in \mathcal{I}} v_t \epsilon_t x_i(t) \geq C\sqrt{d_{\max}R\log(Tn)/|E|}\}$ for some sufficiently large universal constant $C$.                                                                      $\square$

**Lemma B.20.** *Let* $\epsilon_t = y_t - \frac{\exp(\mathbf{x}(t)^\top \boldsymbol{\theta}^*(t))}{1+\exp(\mathbf{x}(t)^\top \boldsymbol{\theta}^*(t))}$. *Under all assumptions in Theorem 3.2, for any fixed integer interval* $\mathcal{I} \subset [1, T]$ *such that* $|\mathcal{I}| \geq c_0(R+1)\frac{|E|}{\lambda_2(L_\mathcal{G})}\log(Tn)$ *for some sufficiently large constant* $c_0 > 0$ *and any fixed* $\mathbf{v} \in D_{\mathcal{I},R}$ *where* $D_{\mathcal{I},R} = \{\mathbf{v} \in \mathbb{R}^{|\mathcal{I}|} : \|D\mathbf{v}\|_0 = R, \min\{k : v_j \neq v_{j+k}\} \geq c_0 \frac{|E|}{\lambda_2(L_\mathcal{G})}\log(Tn)\}$ *with a fixed integer* $R$, *it holds for any* $\kappa > 0$ *that*

$$\max_{i \in [n]} \mathbb{P}\left[|\sum_{t \in \mathcal{I}} v_t \epsilon_t x_i(t)| \geq \kappa\right] \leq \max\{2\exp(-\frac{C|E|\kappa^2}{d_{\max}\sum_{t \in \mathcal{I}} v_t^2}), (Tn)^{-3R-12}\}.$$

*Proof.* Following the same arguments in the proof of Lemma B.21, we have index set of nonzero terms $\mathcal{I}_i$ for each $i \in [n]$. Furthermore, let $\{\mathcal{J}_k\}_{k \in [R+1]}$ be the $R+1$ subintervals such that for each $k$, $v_j$ takes identical values for all $j \in \mathcal{J}_k$. Since $R$ is fixed, by similar arguments we can prove that uniformly for $k \in [R+1]$ and $i \in [n]$, we have $|\mathcal{I}_i \cap \mathcal{J}_k| \leq \frac{3d_{\max}}{|E|}|\mathcal{J}_k|$ with probability at least $1 - (Tn)^{-4R-13}$. Now we condition on this event.

By definition, $\mathbb{E}[\epsilon_t | \mathbf{x}(t)] = 0$, so for each $i \in [n]$, if we let $S_i(t) = \sum_{j \in [t]} v_{l_{i,t}} \epsilon(l_{i,t}) x_i(l_{i,t})$ for $t \in [|\mathcal{I}_i|]$ and $S_i(0) = 0$, then $\{S_i(t)\}$ is a martingale with respect to the filtration $\{\mathcal{F}_t : \mathcal{F}_t = \sigma(\mathbf{x}(l_{i,1}), \cdots, \mathbf{x}(l_{i,t}))\}$. Furthermore, for any $t \in [1, T]$,

$$|S_i(t) - S_i(t-1)| \leq |v_{l_{i,t}} x_i(l_{i,t})| \leq |v_{l_{i,t}}|.$$

Thus by Lemma B.23 we have

$$\mathbb{P}\left[|\sum_{t \in \mathcal{I}} v_t \epsilon_t x_i(t)| \geq \kappa\right] \leq 2\exp(-\frac{C\kappa^2}{\sum_{t \in \mathcal{I}_i} v_t^2}).$$

Now by the fact that $|\mathcal{I}_i \cap \mathcal{J}_k| \leq \frac{3d_{\max}}{|E|}|\mathcal{J}_k|$ for each $i, k$, we have $\sum_{t \in \mathcal{I}_i} v_t^2 \leq \frac{3d_{\max}}{|E|}\sum_{t \in \mathcal{I}} v_t^2$. Then the conclusion follows from a union bound.                                                                      $\square$

**Lemma B.21** (General graph). *Let* $\epsilon_t = y_t - \frac{\exp(\mathbf{x}(t)^\top \boldsymbol{\theta}^*(t))}{1+\exp(\mathbf{x}(t)^\top \boldsymbol{\theta}^*(t))}$. *Under all assumptions above, for any integer interval* $\mathcal{I} \subset [1, T]$ *such that* $|\mathcal{I}| \geq C_0 \frac{|E|}{\lambda_2(L_\mathcal{G})}\log(Tn)$ *for some sufficiently large constant* $C_0 > 0$, *it holds with probability at least* $1 - (Tn)^{-10}$

$$\max_{i \in [n]} |\sum_{t \in \mathcal{I}} \epsilon_t x_i(t)| \leq \sqrt{\frac{d_{\max}}{|E|}|\mathcal{I}|\log(Tn)}.$$

*Proof.* By the assumptions above, i.e., in the comparison graph at each time point a single edge is uniformly randomly picked from the edge set $E$ of $\mathcal{G}$, we know that $\mathbb{P}[|x_i(t)| = 1] = \frac{d_i}{|E|} \leq \frac{d_{\max}}{|E|}$. Therefore, it follows from a Chernoff inequality (Lemma B.22) that for each $i \in [n]$ with probability at least $1 - (Tn)^{-12}$,

$$\sum_{t \in \mathcal{I}} |x_i(t)| - \frac{d_i}{|E|}|\mathcal{I}| \leq c\sqrt{\log(Tn)} \cdot \sqrt{\frac{d_i}{|E|}|\mathcal{I}|}.$$

Since $\lambda_2(L_\mathcal{G}) \leq 2d_{\max}$ and $|\mathcal{I}| \geq C_0 \frac{|E|}{\lambda_2(L_\mathcal{G})} \log(Tn)$, we have $|\mathcal{I}| \geq c_0 \frac{|E|}{d_{\max}} \log(Tn) \geq c_0 \frac{|E|d_i}{d_{\max}^2} \log(Tn)$ and thus

$$\sum_{t \in \mathcal{I}} |x_i(t)| - \frac{d_i}{|E|}|\mathcal{I}| \leq c\sqrt{\log(Tn)} \cdot \sqrt{\frac{d_i}{|E|}|\mathcal{I}|} \leq C\frac{d_{\max}}{|E|}|\mathcal{I}|,$$

which implies that with probaility at least $1 - (Tn)^{-11}$, it holds uniformly for all $i \in [n]$ that in summation $\sum_{t \in \mathcal{I}} \epsilon_t x_i(t)$ there are at most $C|\mathcal{I}|\frac{d_{\max}}{|E|}$ nonzero terms.

Now we condition on this event and denote for each $i \in [n]$ the index set of nonzero terms as $\mathcal{I}_i$. Thus we have $\sum_{t \in \mathcal{I}} \epsilon_t x_i(t) = \sum_{t \in \mathcal{I}_i} \epsilon_t x_i(t)$. For each $\mathcal{I}_i$, we write its elements as $l_{i,t}$ for $t \in [|\mathcal{I}_i|]$ such that $l_{i,1} < l_{i,1} < \cdots < l_{i,|\mathcal{I}_i|}$.

By definition, $\mathbb{E}[\epsilon_t|\mathbf{x}(t)] = 0$, so for each $i \in [n]$, if we let $S_i(t) = \sum_{j \in [t]} \epsilon(l_{i,t}) x_i(l_{i,t})$ for $t \in [|\mathcal{I}_i|]$ and $S_i(0) = 0$, then $\{S_i(t)\}$ is a martingale with respect to the filtration $\{\mathcal{F}_t : \mathcal{F}_t = \sigma(\mathbf{x}(l_{i,1}), \cdots, \mathbf{x}(l_{i,t}))\}$. Furthermore, for any $t \in [1, T]$,

$$|S_i(t) - S_i(t-1)| \leq |x_i(l_{i,t})| \leq 1.$$

Thus by Azuma's inequality (Lemma B.23) and a union bound we can get the conclusion. □

**Lemma B.22.** *Suppose* $Z_1, \cdots, Z_s$ *are independent random variables with zero expectation and variance* $\mathbb{E}Z_i^2 = \sigma_i^2$ *satisfying* $|Z_i| \leq 1$ *almost surely, then*

$$\mathbb{P}\{|\sum_{i \in [s]} Z_i| \geq u\sigma\} \leq C \max\{e^{-cu^2}, e^{-cu\sigma}\},$$

*where* $\sigma^2 = \sum_{i \in [s]} \sigma_i^2$, *and* $C, c > 0$ *are universal constants. In particular, for* $u \leq \sigma$, *we have*

$$\mathbb{P}\{|\sum_{i \in [s]} Z_i| \geq u\sigma\} \leq Ce^{-cu^2}.$$

*Proof.* See Theorem 2.1.3 in Tao (2012). □

A sequence of random variables $\{D_k\}_{k \in \mathbb{Z}_+}$ is called a martingale difference if there exists a martingale $(Z_k, \mathcal{F}_k)_{k \in \mathbb{Z}_+}$ such that $D_k = Z_k - Z_{k-1}$. The following result is well-known in high-dimensional statistics (Wainwright, 2019). We include the proof for completeness and the convenience of readers.

**Lemma B.23** (Azuma's Inequality or Azuma-Hoeffding Inequality). *Suppose* $\{D_k\}_{k \in \mathbb{Z}_+}$ *is a martingale difference. If* $D_k \in (a_k, b_k)$ *almost surely for some* $a_k < b_k$, *then*

$$\mathbb{P}\left(\left|\sum_{k=1}^n D_k\right| \geq t\right) \leq 2\exp\left\{-\frac{2t^2}{\sum_k(b_k - a_k)^2}\right\}. \tag{B.31}$$

*Proof.* $D_k \in (a_k, b_k)$ almost surely implies that for almost all $\omega \in \Omega$, the conditional variable $(D_k|\mathcal{F}_{k-1})(\omega) \in (a_k, b_k)$ almost surely, where $(D_k|\mathcal{F}_{k-1})(\omega)$ is defined using regular conditional distributions. By the Hoeffding's bound, $(D_k|\mathcal{F}_{k-1})(\omega)$ is sub-Gaussian with parameter $\sigma^2 = (b_k - a_k)^2/4$, for almost all $\omega$. Therefore by the definition of sub-Gaussian random variables, we have that for almost all $\omega$,

$$\mathbb{E}\left[\exp\{\lambda(D_k|\mathcal{F}_{k-1})(\omega)\}\right] \leq \exp\left\{\lambda^2\frac{(b_k - a_k)^2}{8}\right\}. \tag{B.32}$$

By the property of regular conditional distributions,

$$\mathbb{E}\left[e^{\lambda D_k}|\mathcal{F}_{k-1}\right](\omega) = \mathbb{E}\left[\exp\{\lambda(D_k|\mathcal{F}_{k-1})(\omega)\}\right], \text{almost surely.} \tag{B.33}$$

Therefore

$$\mathbb{E}\left[e^{\lambda D_k}|\mathcal{F}_{k-1}\right] \leq \exp\left\{\lambda^2 \frac{(b_k - a_k)^2}{8}\right\}, \text{almost surely.} \tag{B.34}$$

Now let $\nu_k^2 = (b_k - a_k)^2/4$ and $\alpha_k = 0$ in Theorem B.24 and we can prove the inequality. $\qquad\square$

A random variable $X$ with $\mathbb{E} = \mu$ is called sub-exponential with parameters $\nu^2$ and $\alpha$, or $\mathrm{SE}(\nu^2, \alpha)$ for brevity, if

$$\mathbb{E}[e^{\lambda(X-\mu)}] \leq e^{\lambda^2\nu^2/2}, \ \forall|\lambda| \leq \frac{1}{\alpha}.$$

**Lemma B.24.** *Let $\{(D_k, \mathcal{F}_k), k \in \mathbb{Z}_+\}$ be a martingale difference s.t.*

$$\mathbb{E}\left[e^{\lambda D_k}|\mathcal{F}_{k-1}\right] \leq e^{\lambda^2\nu_k^2/2}, \ \forall|\lambda| \leq \frac{1}{\alpha_k}, \tag{B.35}$$

*almost surely. Then*

*1)* $\sum_{k=1}^n D_k \in \mathrm{SE}(\sum_k \nu_k^2, \max_k \alpha_k)$;

*2)*

$$\mathbb{P}(|\sum_k D_k| \geq t) \leq \begin{cases} 2\exp\left\{-\frac{t^2}{2\sum_k \nu_k^2}\right\}, \ t \leq \frac{\sum_k \nu_k^2}{\max_k \alpha_k}, \\ 2\exp\left\{-\frac{t}{2\max_k \alpha_k}\right\}, \ t > \frac{\sum_k \nu_k^2}{\max_k \alpha_k}. \end{cases} \tag{B.36}$$

*Proof.* 1). By the iterated law of expectation

$$\begin{aligned} \mathbb{E}\left[e^{\lambda\sum_{k=1}^n D_k}\right] &= \mathbb{E}\left[\mathbb{E}\left[e^{\lambda\sum_{k=1}^n D_k}|\mathcal{F}_{n-1}\right]\right] \\ &= \mathbb{E}\left[\exp\{\lambda\sum_{k=1}^{n-1} D_k\}\mathbb{E}\left[e^{\lambda D_n}|\mathcal{F}_{n-1}\right]\right] \\ &\leq \mathbb{E}\left[\exp\{\lambda\sum_{k=1}^{n-1} D_k\}e^{\lambda^2\nu_n^2/2}\right] \\ &= e^{\lambda^2\nu_n^2/2}\mathbb{E}\left[e^{\lambda\sum_{k=1}^{n-1} D_k}\right], \text{ for}|\lambda| < \frac{1}{\alpha_n}, \end{aligned}$$

where we use the fact that $\exp\{\lambda\sum_{k=1}^{n-1} D_k\} \in \mathcal{F}_{n-1}$ and (B.35). Repeating the same procedure for $k = n-1, \cdots, 2$, we can get

$$\mathbb{E}\left[e^{\lambda\sum_{k=1}^n D_k}\right] \leq e^{\lambda^2\frac{\sum_{k=1}^n \nu_k^2}{2}}, \text{ for}|\lambda| < \frac{1}{\max_k \alpha_k}. \tag{B.37}$$

2) Use the property of sub-exponential random variables and 1). $\qquad\square$