# OpenReview forum: "Detecting Abrupt Changes in Sequential Pairwise Comparison Data"
_NeurIPS.cc/2022/Conference — NeurIPS 2022 Accept_

### Official Review · Reviewer_8mZi · 2022-07-09

**Rating:** 7
**Confidence:** 2
**Soundness:** 4 excellent
**Presentation:** 4 excellent
**Contribution:** 4 excellent

**Summary:**

This paper aims to address the novel problem of estimating multiple change points from pairwise comparison time series, which is modeled using the existing time-varying BTL model.

The comparison pair is assumed to be randomly chosen from n items (Eq. 2.1). The probability of one item beats another item is modeled as a Bernoulli distribution (Eq. 2.2). The parameter estimation can be achieved by minimizing the negative log-likelihood with regularization (Eq. 2.6, Eq. 3.1). Two algorithms (Algorithm 1 dynamic programming and Algorithm 2 local refinement) are proposed. The major contribution of this work is to prove that both algorithms are consistent under assumption 3.1 (Theorem 3.2, Theorem 3.3).

Experiments on simulated data shows the proposed approach can estimate the correct number of change points and the location of those points much more accurately compared to the baseline (Table 1).

**Questions:**

See the weakness section above

**Limitations:**

The author did address the potential negative societal impact of this work.

**Strengths And Weaknesses:**

Strengths
1) The major contribution of this work, i.e., estimating multiple change points from pairwise comparison time series, seems novel.
2) The proposed algorithms have solid theoretical guarantees on the consistency, although the signal to noise ratio assumption might be difficult to check.
3) Experiments on simulated data shows the clear advantage of the proposed approach.
4) The motivation is clearly presented. The related works of BLT model are well introduced.

Weaknesses
1) In the simulated experiments, would it be better if the ground truth parameter \theta^* are generated by random sampling instead of fully determined by the constant \delta?
2) Besides the three types of changes listed in the simulated setting, it would also be useful to see how the algorithm performs under random perturbation on \theta^* and check how the proposed approach performs when the perturbation strength increases.

---

> ### Author Response · Authors · 2022-08-02
> **First response to Reviewer 8mZi**
>
> We thank the reviewer for their careful reading of our manuscript and for their constructive comments. In what follows we reply to the comments and questions point-by-point.
>
> **Reviewer comment 1: Ground truth.** "In the simulated experiments, would it be better if the ground truth parameter $\theta^*$ are generated by random sampling instead of fully determined by the constant $\delta$?"
>
> **Author response:** In the revision, we conduct new experiments where entries of the parameter $\theta^*$ are randomly sampled from the uniform distribution. The simulation setup and the numerical results are summarized in Section A.2 of the appendix. Our proposed method DPLR continues to provide accurate change point estimation even in these more challenging settings where entries of $\theta^*$ are random.
>
> **Reviewer comment 2: Type of changes.** "Besides the three types of changes listed in the simulated setting, it would also be useful to see how the algorithm performs under random perturbation on $\theta^*$ and check how the proposed approach performs when the perturbation strength increases."
>
> **Author response:** In the revision, we conduct additional experiments where the perturbations (changes) of the true signal $\theta^*$ at the change points are random. Specifically, at time 0 we generate random $\theta^*$ from uniform distribution. Then we randomly permute $\alpha$% of the entries of $\theta^*$ at each change point, where $ \alpha \% \in\{50\%, 75\%, 100\%\}$. Note that as the proportion of the randomly permuted entries increases, the random perturbation strength raises at the change points. As shown in Table 4 in Section A.2 of the appendix, our algorithm DPLR is able to provide more accurate change point estimations as the random perturbation strength increases.

---

### Official Review · Reviewer_4GCo · 2022-07-11

**Rating:** 4
**Confidence:** 2
**Soundness:** 2 fair
**Presentation:** 2 fair
**Contribution:** 2 fair

**Summary:**

The paper proposes a dynamic programming based algorithm to localize the change points in a high-dimensional Bradley-Terry-Luce (BTL) model with piece-wise constant parameters. The proposed algorithm is also proven to have consistent estimation of the unknown locations of the change points.

**Questions:**

Could the author/s answer some of my concerns in the above section? Thanks.

**Strengths And Weaknesses:**

Strengths:
  1. The paper considers a new problem setup for the pairwise comparison ranking problem under the piece-wise constant time-varying BTL model.
  2. The paper proposes a dynamic programming based algorithm to locate the change points under the piece-wise constant time-varying BTL model with proven consistent estimation in the change locations.

Weaknesses:
  1. The paper is a bit hard to read/understand. Some terms/equations are not defined adequately. For example, What is the diverging sequence B_T and \Delta_{\kappa} in Eq.(3.3)? How to derive the equality in Eq.(3.8)? What is F(I) in Algorithm 1for loop?
  2. The assumption of the distinct pair of items to be compared with at each time step being randomly chosen from the whole items and independently over time is very restrictive. For example, I don't think the pairs from the practical example used in the paper's Experiment section is randomly chosen and independently over time.
  3. I am concerned with some of the statements regarding the Remark 2 on the sharpness of the signal-to-noise ratio condition and the statement for the Theorem 3.3. The author/s explained a lot in why it makes sense to be in the form of Eq.(3.3) for most parameters except the minimal winning probability p_{lb}. But p_{lb} is one of the most important parameters in this case and the explanation seems a bit weak to me. For the Theorem 3.3, the author/s conjecture that it's minimax optimal. But for a theoretical paper, it's not rigorous since for the problem setup in this paper, it assumes that the comparison data comes from some distributions. Then in what sense of minimax it is optimal? Is there any condition for that optimality?
  4. I am also concerned with the computation complexity of the proposed algorithm, which is O(T^2C(T)), where C(T) is the complexity of solving min_{\theta} L(\theta, [1,T]). Since this algorithm targets the problem setup of the sequential pairwise comparison, the time horizon T is usually very large which makes the computation very slow.

---

> ### Author Response · Authors · 2022-08-02
> **First response to Reviewer 4GCo (Part 2)**
>
> We thank the reviewer for their careful reading of our manuscript and for their constructive comments. Since replying to all the four comments by the reviewer leads to an excess of the character limit of a single response, we make our response into two parts (this is part 2). We apologize for any inconvenience in reading the response.
>
> **Reviewer comment 3:** "I am concerned with some of the statements regarding the Remark 2 on the sharpness ... Is there any condition for that optimality?"
>
> **Author response:** As we mentioned in our paper (line 102-106), while $p_\{lb\}$ is a very important parameter, it is almost always assumed to be a constant in the ranking literature (Shah et al. 2016; Chen et al. 2022). The fact that we allow for $p_\{lb\}$ to change and that we make the dependence on $p_\{lb\}$ explicit in our rates is already an improvement over most of the literature on the BLT model. In fact, explicitly matching $p_\{lb\}$ in the minimax rate of estimating the model parameters is an unsolved problem in the ranking literature (Negahban et al. 2017) and it goes beyond the scope of our paper. On the other hand, as we mentioned in Remark 2, in the single-change-point setting, our SNR condition does match the two-sample testing bound in the $p_\{lb\}$ parameter. We do agree with the reviewer that it would be very desirable to characterize the dependence of the localizaiton error on $p_\{lb\}$ and we intend to pursue this line of research in future work.
>
> We also thank the reviewer for their comments about minimax optimality. In our setting, minimax optimality is in the sense that it covers the class of joint distributions of $\\{y_t, x(t)\\} \subset \\{0, 1\\}\times \\{−1, 0, 1\\}^n$, where $x(t)$ is a independently selected pair and $y_t$ is the outcome of the comparison given $x(t)$, which is defined in Eq.(2.5) as
> $$P_\{\theta^*(t)\}[y_t = 1|x(t)] = \psi(x(t)^\{\top\}\theta^*(t)).$$
> It is well known that the BLT model can be viewed as a high-dimensional variant of the logistic regression model in the ranking settings in which the covariates take values in $\\{0, −1, 1\\}^n$. In the modern statistical literature, for both linear regression and generalized linear regression models, the covariates $\{x(t)\}$ are usually assumed to be random. This assumption does not conflict with the definition, analysis or the conclusion of the minimax optimality in the linear regression or generalized linear regression models (Raskutti et al. 2011; Negahban et al. 2012; Rinaldo et al. 2021).
>
> **Reviewer comment 4: Computation complexity**
>
> **Author response:** Thank you for your comments on computational complexity. We would like to point out that due to the non-convexity of $\ell_0$-norm, directly solving the optimization in Eq. (3.1) would take $O(T^\{K_\{max\}\} C(T ))$ many operations, where $K_\{max\}$ is a user-specified upper bound for $K$, the number of true change points. With Dynamic Programming (DP), the complexity can be greatly reduced to $O(T^2C(T ))$.
>
> We would like to emphasize that our paper proposes a new change point ranking problem and develops a new methodology to tackle this problem. To date, there is no other existing algorithm that can consistently detect change points in the sequential paired comparison time series data. So our method is state-of-the-art in terms of both computational and statistical efficiency. With that being said, it comes to our attention that there is a recent work (Xu et al. 2022) on change point detection which shows that, with proper implementation, the computation complexity of DP can be reduced to $O(T^2C(n))$, where $n$ is the number of items to be ranked, and $n$ is usually negligible compared to $T$. Our method can also achieve the same order of complexity with this new procedure incorporated.
>
>
> References
>
> Chen et al. (2022). Partial Recovery for Top-K Ranking: Optimality of MLE and Sub-Optimality of Spectral Method. To appear in The Annals of Statistics.
>
> Negahban et al. (2017). Rank Centrality: Ranking from Pairwise Comparisons. Oper. Res.
>
> Negahban et al. (2012). A unified framework for high-dimensional analysis of M-estimators with decomposable regularizers. Statistical Science.
>
> Raskutti et al. (2011). Minimax rates of estimation for high-dimensional linear regression over lq-balls. IEEE transactions on information theory.
>
> Rinaldo et al. (2021). Localizing changes in high-dimensional regression models. AISTATS 2021.
>
> Shah et al. (2016). Estimation from Pairwise Comparisons: Sharp Minimax Bounds with Topology Dependence. JMLR.
>
> Xu et al. (2022). Change point inference in high-dimensional regression models under temporal dependence. arXiv:2207.12453.

---

> ### Author Response · Authors · 2022-08-02
> **First response to Reviewer 4GCo (Part 1)**
>
> We thank the reviewer for their careful reading of our manuscript and for their constructive comments. Since replying to all the four comments by the reviewer leads to an excess of the character limit of a single response, we make our response into two parts (this is part 1). We apologize for any inconvenience in reading the response.
>
> **Reviewer comment 1: Presentation of the paper.**
>
> **Author response:**
> For the diverging sequence in Eq. (3.3), we meant any sequence $\\{B_T\\}$ diverging to infinity as $T$ goes to infinity. Here the divergence rate of $B_T$ can be arbitrarily slow, e.g., it can be of order $\log \log(T)$. We would like to point out that this is a standard way of formulating signal-to-noise ratio conditions that guarantee consistency of the change point detection procedure, and can be found in many contributions in the time series literature.
>
> We would like to clarify that the term $\Delta \kappa^2$ in Eq. (3.3) means the product of $\Delta$ and $\kappa^2$ but not $\Delta$ with subscript $\kappa$. Here, $\Delta$ denotes the minimal spacing between consecutive change points and $\kappa$ denotes the minimal jump size at the change points. See Eq. (2.7) of our paper for more precise definitions.
>
> Eq. (3.8) follows since
> $$ \max_\{k \in [K] \} | \widetilde \eta_k -\eta_k |  \leq  C_P \frac{p_\{l b\}^\{-4\} K n^2 \log (T n)}{\kappa^2}\leq C_P\frac{\Delta}{B_\{T\}} = o(\Delta),$$
> where we use the SNR assumption $\Delta\cdot \kappa^{2} \geq B_T p_\{l b\}^\{-4\} K n^2 \log (T n)$ in the last inequality and the fact that $B_T$ diverges in the final step. We have clarified this point in the revision.
>
> There is indeed a typo in Algorithm 1: we used $\mathcal{F}(\mathcal{I})$ to represent the goodness-of-fit function and here it should be $L(\hat{\theta}(\mathcal{I}), \mathcal{I})$, the fitted negative log-likelihood on the interval $\mathcal{I}$, and we have fixed it in the revision.
>
> **Reviewer comment 2:** "The assumption of the distinct pair of items ... the practical example used in the paper’s Experiment section is randomly chosen and independently over time."
>
> **Author response:**
> We completely agree with the reviewer that it would be helpful and interesting to allow for temporal dependence in our model settings. We believe that, using the same techniques as in Xu et al. 2022, our results continue to hold even when the randomly selected pairs $x(t)$ are temporally correlated.
>
> With that said, our assumptions are no stronger than those routinely made in the ranking literature, even in the most recent and sophisticated contributions. The assumption that the compared pairs are randomly and independently selected, though restrictive, is widely used in the theoretical statistical ranking literature (Shah et al. 2016; Negahban et al. 2017; Chen et al. 2022). Moreover, even in the sequential setting, independence of the samples across time is a frequently used assumption in change point detection literature (Frick et al. 2014; Wang and Samworth 2018).
>
> We would also like to explain the independently selected pairs assumption from a practical perspective. Games in basketball or baseball for examples, are usually arranged in a way so that teams are involved in a fairly even manner. For instance, it cannot happen that a specific team does not play any game for half of the season. Therefore, with enough teams in a sport league, the opponents of each team is relatively randomly selected over the season.
>
> References
>
> Chen et al. (2022). Partial Recovery for Top-K Ranking: Optimality of MLE and Sub-Optimality of Spectral Method. To appear in The Annals of Statistics.
>
> Frick et al. (2014). Multiscale change point inference. JRSSB.
>
> Negahban et al. (2017). Rank Centrality: Ranking from Pairwise Comparisons. Oper. Res.
>
> Shah et al. (2016). Estimation from Pairwise Comparisons: Sharp Minimax Bounds with Topology Dependence. JMLR.
>
> Wang and Samworth (2018). High dimensional change point estimation via sparse projection. JRSSB.
>
> Xu et al. (2022). Change point inference in high-dimensional regression models under temporal dependence. arXiv:2207.12453.

---

### Official Review · Reviewer_4hC2 · 2022-07-12

**Rating:** 6
**Confidence:** 3
**Soundness:** 3 good
**Presentation:** 4 excellent
**Contribution:** 2 fair

**Summary:**

The paper considers the problem of detecting changes in the parameters of the Bradley-Terry-Luce (BTL) model using sequential data on pairwise comparisons among items. The model assumes that the parameters are piece-wise constant, and the goal is to detect the change-points given an entire time-series where at each time step a comparison between pairs of items chosen at random is performed. The paper proposes two novel algorithms (a basic one and a more refined one) that build on standard dynamic programming approaches. The paper proves theoretical guarantees for the proposed algorithms, deriving finite sample error rates that depend explicitly on the several parameters of the model. The refined algorithm is tested and compared with a “potential” competitor (since no other algorithm is available for the same problem) on simulated data. The refined algorithm is also tested on real sports data (i.e., NBA games).

**Questions:**

Have you run the “potential” competitor on the NBA data? If yes, what are the difference in the results with the ones obtained with your method?

**Limitations:**

Yes (at the end of Section 6)

**Strengths And Weaknesses:**

Strenghts:
- The sequential, non-stationary scenario is original, highly significant and of practical importance
- The main contributions are the problem definition, its formulation, and the analysis of the proposed algorithms. (The algorithms seem to fall into a fairly standard approach for similar problems.)
- The paper is clearly written, the presentation is of high quality
- The results on simulated data show that the method works well on data from the assumed model
- The results on sports data seem interesting and in agreement with domain knowledge

Weaknesses:
- The BTL model seems to be less relevant and less studied than other ranking models (e.g., the Mallows model, mentioned by the authors)
- There is no comparison with the “potential competitor” on real data. Since the simulated data comes from the model assumed by the proposed algorithms, it is not surprising that the proposed method works better than the potential competitor. It is therefore important to compare the methods on real data.
- Algorithm 1 is poorly formatted and takes a lot of space (e.g., the if-then statement can be inline, there is no need for the “end for”, “end if”, etc.)

---

> ### Author Response · Authors · 2022-08-02
> **First response to Reviewer 4hC2**
>
> We thank the reviewer for their careful reading of our manuscript and for their constructive comments. In what follows we reply to the comments and questions point-by-point.
>
> **Reviewer comment 1: Model.** "The BTL model seems to be less relevant and less studied than other ranking models (e.g., the Mallows model, mentioned by the authors)."
>
> **Author response:** We agree that the Mallows model is quite popular in statistics and machine learning literature. The BTL model is also widely used to analyze pairwise comparison data in practice, due to its clear form, high interpretability and nice theoretical and computational properties. In fact, over the last few years the statistics and machine learning literature has witnessed a flurry of work and of novel and exiting results (most of which referenced in our manuscript) about the behavior and applicability of the BLT model in high-dimensional settings. See for example Chen et al. 2019, Hendrickx et al. 2020, Jin et al. 2020, Karle and Tyagi 2021 and Chen et al. 2022. Our contributions fit naturally within this modern line of research. Therefore, we believe that investigating the expressive power and practicality of the BTL model in a change point framework is not only interesting in its own right (because it has never been done before) but it can also provide the tools and insight for considering more complex ranking models.
>
> **Reviewer comment 2: Comparison.**
>
> **Author response:** Thank you for suggesting a comparison between our procedure and the “potential competitor” in our real data study. We did not initially pursue this comparison because the former methodology, though sensible, is also completely new and has not been investigated theoretically. Indeed, it is worth emphasizing that, to the best of our knowledge, the problem and settings considered in our submission have never been considered in the literature before and, as a result, there are just no competing methods to ours at all.
>
> In the revision, we have applied the “potential competitor” WBS-GLR to the NBA dataset. All of our findings and conclusions are fully discussed in Section A.3.2 of the appendix. In particular, we include a detailed discussion on the tuning parameter selection of WBS-GLR as well as the results of WBS-GLR in this real data study. We provide a brief summary here for your convenience. As demonstrated in Table 6 of the appendix, WBS-GLR is able to detect several important change points in the NBA history, e.g., the dominance of Celtics and Lakers in 1980s, the Bulls dynasty in 1990s, and the rise of Spurs afterwards. However, compared with DPLR, WBS-GLR fails to detect the rise of Heat and Warriors. Therefore, the outcome of DPLR is more informative in this real application, which again conforms with our findings in the simulation study in Section 4 of our paper.
>
>
> **Reviewer comment 3: Algorithm 1.** "Algorithm 1 is poorly formatted and takes a lot of space (e.g., the if-then statement can be inline, there is no need for the “end for”, “end if”, etc.)"
>
> **Author response:** We have reformatted Algorithm 1 in the revision. It is now presented in a more compact way.
>
>
> **Reviewer Question:** Have you run the “potential” competitor on the NBA data? If yes, what are the difference in the results with the ones obtained with your method?
>
> **Author response:** Please see our reply to your second comment.
>
> **References**
>
> Chen, Pinhan et al. (2022). Partial Recovery for Top-K Ranking: Optimality of MLE and Sub-Optimality of Spectral Method. To appear in The Annals of Statistics.
>
> Chen, Yuxin et al. (2019). Spectral Method and Regularized MLE are Both Optimal for Top-K Ranking. The Annals of Statistics.
>
> Hendrickx et al. (2020). Minimax Rate for Learning From Pairwise Comparisons in the BTL Model. ICML 2020.
>
> Jin, Tao et al. (2020). Rank Aggregation via Heterogeneous Thurstone Preference Models. AAAI 2020.
>
> Karle, Eglantine and Tyagi, Hemant (2021). Dynamic Ranking with the BTL Model: A Nearest Neighbor based Rank Centrality Method. arXiv:2109.13743.

---

### Author Response · Authors · 2022-08-02
**General response to all reviewers**

We are very grateful to all three reviewers for their helpful reviews and constructive comments. In our revision, we have responded to all the comments and questions point-by-point and conducted additional experiments on simulated and real data to address these comments and questions.

We would also like to emphasize that one main contribution of our paper is the proposal of a consistency-guaranteed methodology that can deal with sequential ranking data with potential change points. This important practical problem has not been tackled by any existing works. Our theoretical findings are confirmed by the additional extensive numerical studies suggested by the reviewers.

---

### Meta-Review · Area_Chair_FR7A · 2022-08-24

**Recommendation:** Accept
**Confidence:** Certain

**Metareview:**

In *Detecting Abrupt Changes in Sequential Pairwise Comparison Data* the authors consider the problem of detecting multiple change points in sequential pairwise comparison data.  The reviewers in general consider the model both well-motivated and novel.  For these reasons I recommend that this paper be accepted.


**Award:**

No

---

### Decision · Program_Chairs · 2022-09-14

Accept